# Tight Bounds On The Distortion of Randomized and Deterministic Distributed Voting

**MohammadAli Abam**[*]
abam@sharif.edu

**Davoud Kareshki**[*]
davood.kareshki12@sharif.edu

**Marzieh Nilipour**[*]
marzie.nilipour@sharif.edu

**MohammadHossein Paydar**[*]
mohammad.pay@sharif.edu

**Masoud Seddighin**[†]
m.seddighin@teias.institute

## Abstract

We study metric distortion in distributed voting, where $n$ voters are partitioned into $k$ groups, each selecting a local representative, and a final winner is chosen from these representatives (or from the entire set of candidates). This setting models systems like U.S. presidential elections, where state-level decisions determine the national outcome. We focus on four cost objectives from Anshelevich *et al.* [1]: avg-avg, avg-max, max-avg, and max-max. We present improved distortion bounds for both deterministic and randomized mechanisms, offering a near-complete characterization of distortion in this model.

For deterministic mechanisms, we reduce the upper bound for avg-max from $11$ to $7$, establish a tight lower bound of $5$ for max-avg (improving on $2 + \sqrt{5}$), and tighten the upper bound for max-max from $5$ to $3$. For randomized mechanisms, we consider two settings: (i) only the second stage is randomized, and (ii) both stages may be randomized. In case (i), we prove tight bounds: $5-2/k$ for avg-avg, $3$ for avg-max and max-max, and $5$ for max-avg. In case (ii), we show tight bounds of $3$ for max-avg and max-max, and nearly tight bounds for avg-avg and avg-max within $[3-2/n,\ 3-2/(kn^*)]$ and $[3-2/n,\ 3]$, respectively, where $n^*$ denotes the largest group size.

## 1 Introduction

In social choice theory, a voting rule is a function that takes agents' preferences over alternatives and selects one as the final outcome. Preferences are usually represented as ranked lists, and the goal is to design a voting rule that best reflects these preferences.

How can we evaluate whether a voting rule is appropriate? There are both *axiomatic* and *quantitative* benchmarks for assessing outcomes [2, 3, 4]. In this paper, we focus on one of the most prominent quantitative measures: *distortion*. The idea is simple: each agent has hidden numerical values—either *costs* or *utilities*—for the alternatives, and their ordinal rankings reflect these values. Suppose our ultimate goal is to optimize an objective function $\phi$, such as social cost, maximum cost, or total utility, based on these hidden values. However, since the voting rule only has access to the agents' ordinal preferences—not the numerical values—it may select a suboptimal outcome with respect to $\phi$. The

---

[*]Computer Engineering Department, Sharif University of Technology, Tehran, Iran

[†]Computer Science Department, Tehran Institute for Advanced Studies (TeIAS), Tehran, Iran

39th Conference on Neural Information Processing Systems (NeurIPS 2025).

| Cost objective | General metric | Line metric |
| --- | --- | --- |
| avg-avg | $[7, 11]$ [1] | $7$ [1] |
| avg-max | $[2 + \sqrt{5}, 11]$ [1] | $2 + \sqrt{5}$ [25] |
| max-avg | $[2 + \sqrt{5}, 5]$ [1] | $2 + \sqrt{5}$ [25] |
| max-max | $[3, 5]$ [1] | $3$ [1] |

Table 1: Known results for deterministic distributed mechanisms under various cost objectives.

distortion of a voting rule captures how far its chosen outcome can be from the optimal one in the worst case. It is defined as the ratio between the value of $\phi$ for the selected outcome and the value of $\phi$ for the optimal alternative in the worst case.

Since its introduction by Procaccia and Rosenschein [4], distortion has been a consistent focus of investigation—not only in voting, but also in related social choice problems such as facility location [5, 6, 7, 8] and matching [7, 9, 10, 11]. Still, the core of the literature lies in voting, with particularly rich results when costs of agents form a metric space [10, 12, 13, 14, 15, 16, 17, 18, 19, 20, 21]. For a comprehensive overview of distortion in voting, we refer to the survey by Anshelevich *et al.* [22].

In this paper, we study the distortion in the metric setting when the voting process is distributed. Unlike centralized voting, in many large-scale scenarios, outcomes emerge via a two-stage manner: decisions are made locally within separate groups of agents, the local outcomes are then aggregated into a final outcome. A notable example is the U.S. presidential election, where each state selects a winner, and the national outcome is determined by a weighted aggregation of the state-level results. More formally, a distributed voting mechanism is a pair $(f_{in}, f_{ov})$, where

1. $f_{in}$ is an *in-group* voting rule that selects a local winner for each group based solely on the preferences of agents within that group.

2. Assuming $R$ is the set of local winners, $f_{ov}$ is an *over-group* voting rule that selects the final winner based on the preferences of $R$ over all alternatives [1] or local winners [23].

The study of distortion in distributed voting was pioneered by Filos-Ratsikas *et al.* [24], who extended the notion of distortion to the utility-based distributed scenario. Later, Anshelevich *et al.* [1] investigated distributed voting in the metric cost setting. In the context of distributed voting, since decisions occur in two stages, it is natural to define separate cost objectives for each level. Building on this, Anshelevich *et al.* [1] introduced four standard objectives combining *average* and *maximum* costs within and across the groups: avg-avg, max-avg, avg-max, and max-max. In the deterministic setting, they proved constant upper and lower bounds for all objectives in both general and line metric spaces, summarized in Table 1. Later, Voudouris [25] focused on the line metric and proposed two simple mechanisms for the avg-max and max-avg objectives. These mechanisms achieve an upper bound of $2 + \sqrt{5}$, closing the corresponding gap derived by [1].

As shown in Table 1, distributed voting on the line metric is well-understood, with tight distortion bounds already achieved. We therefore turn to general metric spaces and explore whether randomization can also improve distortion in the distributed setting. This work presents the first formal investigation into randomized distributed mechanisms within the metric setting.

In this paper, we make significant progress on the distortion of distributed voting mechanisms in two main directions. First, we improve the existing distortion bounds of *deterministic* mechanisms with respect to the avg-max, max-avg, and max-max objectives. Second, we explore rules that incorporate *randomized* mechanisms—either in the second stage only, or in both stages—referred to as rand-det and rand-rand, respectively. The output of a randomized mechanism is a probability distribution over the alternatives, rather than a single winner. For both the rand-det and rand-rand mechanisms, we prove tight bounds for almost all of the objectives.

## 1.1 Further Related Work

The most relevant studies to our work [1, 25] are discussed in Section 1. Here, we briefly review other related studies. Since Procaccia and Rosenschein's seminal work [4], research on distortion in

social choice problems has expanded, covering utilitarian settings [26, 27, 28, 29, 30], metric settings [12, 14, 31, 32, 16, 17, 33, 20], and combined approaches [34].

In deterministic case, Anshelevich *et al.* [12] pioneered the study of distortion for the metric framework. Using a simple example, they show that the distortion of any deterministic voting rule is at least 3. Gkatzelis *et al.* [14] proposed an elegant and intricate voting rule, *Plurality Matching*, which achieves a tight distortion of 3. Next, Kizilkaya and Kempe [32] attained the same upper bound with a simpler voting rule, *Plurality Veto*. Filos-Ratsikas *et al.* [24] pioneered distortion analysis in distributed voting under the utilitarian framework. Their work extended to other social choice problems, including facility location [35], aiming to select a single location from a set of alternatives. More recently, Voudouris [36] investigated the distributed distortion in obnoxious voting, where alternatives are undesirable.

Unlike deterministic voting rules, randomized rules can achieve distortion below 3. Anshelevich and Postl [15] proved that the metric distortion of *Random Dictatorship* is at most $3 - 2/n$, with $n$ agents, and establish a lower bound of 2 for any randomized voting rule. Kempe [37] improved the upper bound for *Random Dictatorship* to $3 - 2/m$, where $m$ is the number of candidates. Charikar and Ramakrishnan [16] further raised the lower bound for any randomized rule to $2.112$. Recently, Charikar *et al.* [33] reduced the upper bound to $2.753$. In the context of distributed voting, Filos-Ratsikas and Voudouris [23] investigated randomized mechanisms under the utilitarian framework, establishing distortion bounds in various cases.

## 1.2 Our Contributions

Our results provide improved upper and lower bounds on the distortion of distributed mechanisms across various combinations of deterministic and randomized voting rules and cost objectives. As summarized in Table 2, most of our bounds are tight—despite the fact that our proposed mechanisms are simple. In addition to general metric spaces, we also analyze the Euclidean setting and derive corresponding bounds under this restriction.

**Randomized Mechanisms.** Previous work on metric distortion in the distributed setting has focused exclusively on deterministic voting rules [1, 25, 38, 36]. In this paper, we take a significant step toward understanding randomized mechanisms in distributed voting. We study two natural classes of randomized mechanisms—rand-det and rand-rand—within general metric spaces, and analyze their performance with respect to the four objectives. See Table 2 for an overview of our results.

rand-det mechanisms, defined as pairs $(f_{in}, f_{ov})$, where $f_{in}$ is a deterministic voting rule and $f_{ov}$ is a randomized one. We derive tight distortion bounds with respect to the all objectives in Section 3.

- **max-max, avg-max:** For both objectives, we derive a tight distortion bound of 3. The lower bound is established through a basic example within a single group on a line metric, simplifying the max-max and avg-max objectives to max. The upper bound is proven by a distributed mechanism that first selects a representative for each group with the *Plurality Matching* rule and then chooses the final winner uniformly at random.

- **max-avg:** We establish a tight distortion bound of 5. The lower bound is proven using a line metric and a novel tool we introduce, called the *Bias Tournament*, which may be of independent interest. For the upper bound, we show that applying a deterministic in-group rule with a distortion at most $\alpha \geq 3$, followed by the *Random Dictatorship* rule , achieves an overall distortion of at most $\alpha + 2$. Since the best achievable value of $\alpha$ is 3 (via the *Plurality Matching* rule), this yields a matching upper bound of 5.

- **avg-avg:** We prove a tight distortion bound of $5 - 2/k$. Obtaining this bound for the avg-avg objective is the most challenging aspect of the rand-det mechanisms. The lower bound construction, though similar to that of the max-avg objective, requires a more delicate analysis to extract the $2/k$ improvement. Once again, we employ the Bias Tournament and model the metric space via shortest-path distances in a graph.

In Section 4, we analyze rand-rand mechanisms, defined as pairs $(f_{in}, f_{ov})$ comprising of two randomized voting rules, and derive tight or near-tight distortion bounds. All the upper bounds are obtained via a distributed mechanism that initially applies the *Random Dictatorship* rule within

| | Objective | Distortion | |
|---|---|---|---|
| | | lower bound | upper bound |
| **det-det** | avg-avg | 7 [1] | 11 [1] |
| | avg-max | $2 + \sqrt{5}$ [1] | 7 (Corollary 5.3) |
| | max-avg | 5 (Theorem 5.4) | 5 [1] |
| | max-max | 3 [1] | 3 (Theorem 5.1) |
| **rand-det** | avg-avg | $5 - \frac{2}{k}$ (Theorem 3.8) | $5 - \frac{2}{k}$ (Corollary 3.4) |
| | avg-max | 3 (Theorem 3.6) | 3 (Theorem 3.5) |
| | max-avg | 5 (Theorem 3.7) | 5 (Corollary 3.2) |
| | max-max | 3 (Theorem 3.6) | 3 (Theorem 3.5) |
| **rand-rand** | avg-avg | $3 - \frac{2}{n}$ (Theorem 4.9) | $3 - \frac{2}{kn^*}$ (Theorem 4.4) |
| | avg-max | $3 - \frac{2}{n}$ (Theorem 4.7) | 3 (Theorem 4.2) |
| | max-avg | 3 (Theorem 4.6) | 3 (Theorem 4.3) |
| | max-max | 3 (Theorem 4.6) | 3 (Theorem 4.1) |

Table 2: An overview of our results for various cost objectives in general metric spaces, with gray-text results indicating those derived from prior work. $n^*$ denotes the size of the largest group. Thus, the bound of $3 - 2/n$ for the avg-avg objective in rand-rand is tight when all group sizes are equal.

each group and then randomly selects the final winner from the chosen representatives with uniform probability.

- **max-max, max-avg:** For both objectives, we establish a tight distortion bound of 3. We construct a shared example consisting of $k$ single-voter groups to establish the lower bound, even when the metric space is a line. In this scenario, the max-max and max-avg objectives both simplify to the max objective.

- **avg-max:** We establish a lower bound of $3 - \frac{2}{n}$, which nearly matches our upper bound of 3. The lower bound is proven with an instance where the number of candidates and voters are equal ($n = m$) and there is only a single group ($k = 1$). Additionally, we conclude a lower bound for the max objective in the centralized setting: we show that any randomized rule must have a distortion of at least $3 - \varepsilon$ for any constant $\varepsilon > 0$. This is particularly interesting since even deterministic rules are known to have an upper bound of 3 for max [14].

- **avg-avg:** We establish a nearly tight distortion bound slightly below 3. For an instance with $k$ single-voter groups on a tree graph, we prove lower bound of $3 - \frac{2}{n}$. We further derive an upper bound of $3 - \frac{2}{kn^*}$, where $n^*$ denotes the largest group size. When all groups are of equal size, it yields matching upper and lower bounds. Notably, deriving these bounds is the most challenging aspect of analyzing rand-rand, mechanisms.

**Deterministic Mechanisms.** We consider det-det mechanisms, defined as pairs $(\mathsf{f}_{in}, \mathsf{f}_{ov})$ comprising of two independently deterministic voting rules, in Section 5. We resolve the previously known gaps for the max-avg and max-max objectives and provide an enhanced upper bound for the avg-max objective. In this section, we adopt a setting akin to [1], where $\mathsf{f}_{ov}$ selects a winner from the set of *all candidates*, not solely those chosen in the first stage.

- **avg-max.** We improve the upper bound from **11** to **7**. Anshelevich *et al.* [1] show that combining the in-group and over-group voting rules with distortions $\alpha$ and $\beta$, respectively, yields an overall distortion of $\alpha + \beta + \alpha\beta$. With their best known values ($\alpha = 3$, $\beta = 2$), this gives 11. We prove that if the in-group rule, $\mathsf{f}_{in}$, merely satisfies the property of *Pareto efficiency*, then the overall distortion is at most $2\beta + 3$, which is *independent* of $\alpha$. This results in a tighter upper bound of 7, and shows the dominant role of the over-group rule.

- **max-avg.** We improve the lower bound from $2 + \sqrt{5}$ to **5**. Interestingly, our lower-bound instance is based on a metric constructed via shortest-path in a graph, rather than a line or Euclidean. This confirms that the upper bound from [1] is indeed tight.

- **max-max.** We improve the upper bound on distortion from **5** to **3** for general metric spaces. Although a bound of 3 was previously known for the *line metric*, the general case remained open. We show that a distributed mechanism same as the *Arbitrary Dictator*, proposed by [1], actually achieves distortion 3 for any metric space.

**Bias Tournament.** The Bias Tournament is a directed graph with one node per candidate. For any pair of candidates $c_1$ and $c_2$, we add a directed edge from $c_1$ to $c_2$ if, in a group containing only two voters with preferences $(c_1, c_2, \ldots)$ and $(c_2, c_1, \ldots)$—where the remaining candidates are ordered identically according to a fixed permutation $\sigma$ over all candidates—the in-group rule $\mathsf{f}_{in}$ deterministically selects $c_1$ as the winner. This construction captures the bias in $\mathsf{f}_{in}$'s tie-breaking behavior across candidate pairs. To analyze the implications of these biases, we use a well-known property of tournaments: any tournament on $m$ nodes contains at least one node with in-degree at least $\lceil (m-1)/2 \rceil$. This fact allows us to identify a *losing* candidate—one frequently defeated in pairwise comparisons—and use it as the basis for constructing an instance with high distortion, which helps us to establish the lower bounds for the rand-det and det-det mechanisms.

**Centralized Setting.** Interestingly, one of our lower bound constructions, originally developed for distributed mechanisms also applies to randomized centralized setting. In particular, we analyze the $\max$ cost objective in this setting. Previously, Gkatzelis *et al.* [14] established an upper bound of 3 for $\max$ using *Plurality Matching*, as defined in Section 2. In our work, we revisit the avg-max objective for the rand-rand mechanisms and prove a distortion lower bound. We then reuse the same construction to show that the distortion of $\max$ under centralized voting is at least $3 - \varepsilon$ for any constant $\varepsilon > 0$, thus matching the known upper bound up to an arbitrarily small gap.

**Euclidean Space.** Our lower-bound constructions for the avg-avg objective rely on general metric spaces that do not reduce to simpler structures such as the line or Euclidean space. This raises the natural question of whether similarly strong bounds can be achieved in more structured settings. In Section 6, we address this by constructing novel instances within a Euclidean hyper-simplex, obtaining lower bounds of $\sqrt{5} - \varepsilon$ for the rand-rand mechanisms and $2 + \sqrt{5} - \varepsilon$ for the rand-det ones, where $\varepsilon$ is arbitrarily small. Specifically, for rand-rand, we fix an integer $l$ and construct an instance with $l + 2$ candidates and $k = l + 1$ single-voter groups embedded in $\mathbb{R}^{l+1}$. Similarly, for rand-det, we construct an instance with $2m$ candidates and $m$ groups of two voters each, embedded in $\mathbb{R}^{m+1}$ (where $m$ is fixed).

## 2 Basic Notations

An instance of the *distributed voting problem* is a tuple $(\mathcal{V}, \mathcal{C}, \mathcal{G}, \pi, \mathsf{d})$. Here, $\mathcal{V}$ is a set of $n$ voters, and $\mathcal{C}$ is a set of $m$ candidates. We use $v_i$ to denote the $i$-th voter and symbols $a$, $b$, $c$, $\mathsf{o}$, and $\mathsf{w}$ to refer to candidates. In particular, $\mathsf{w}$ typically denotes the winner and $\mathsf{o}$ refers to the optimal candidate. $\mathcal{G}$ is a partition of voters into $k$ groups, such that each voter belongs to exactly one group. For each group $g \in \mathcal{G}$, $n_g$ is the number of voters in $g$. We denote the optimal candidate for group $g$ by $\mathsf{o}_g$. We say an instance is *symmetric*, if all groups have equal sizes. Each voter $v_i$ has a strict ranking $\pi_i$ over the set of candidates, representing their ordinal preferences. A preference profile is the collection of preferences from all voters, denoted by $\pi = (\pi_1, \ldots, \pi_n)$. These preferences arise from a cost function based on the underlying metric space $\mathsf{d}$. For each voter $v$ and candidate $c$, $\mathsf{d}(v, c)$ denotes the cost of candidate $c$ for voter $v$. The voters rank all candidates in increasing order of cost, which means they prefer those who are closer. The distance function $\mathsf{d}$ satisfies the standard metric properties of *non-negativity*, *symmetry*, and the *triangle inequality*.

We denote the top-ranked candidate of each voter $v$ as $\mathsf{top}(v)$. For a candidate $c$ and a group $g \in \mathcal{G}$, $v^*(c, g) = \arg\max_{v \in g} \mathsf{d}(v, c)$ denotes the farthest voter from $c$ within group $g$. Similarly, $v^{**}(c) = \arg\max_{v \in \mathcal{V}} \mathsf{d}(v, c)$ represents the farthest voter from $c$ across all voters. We also use $\mathsf{cost}_g(c)$ to denote the cost of candidate $c$ restricted to group $g$, and define $g^*(c) = \arg\max_{g \in \mathcal{G}} (1/n_g) \sum_{v \in g} \mathsf{d}(v, c)$ that is, the group in which candidate $c$ incurs the highest average cost.

Given an instance $\mathcal{I}$, various cost objectives can be considered to evaluate the final winner. In the distributed voting, we can be even more flexible by applying different objectives at each stage. Following [1], we consider four cost objectives:

- **Average of averages** (avg-avg): Average the costs within each group, then average across all groups; $\mathsf{avg\text{-}avg}(c \mid \mathcal{I}) = \frac{1}{k} \sum_{g \in \mathcal{G}} \frac{1}{n_g} \sum_{v \in g} \mathsf{d}(v, c) = \frac{1}{k} \sum_{g \in \mathcal{G}} \mathsf{cost}_g(c)$.

- **Average of maxima** (avg-max): Find the most dissatisfied voter in each group, then average their costs; $\mathsf{avg\text{-}max}(c \mid \mathcal{I}) = \frac{1}{k} \sum_{g \in \mathcal{G}} \max_{v \in g} \mathsf{d}(v, c) = \frac{1}{k} \sum_{g \in \mathcal{G}} \mathsf{d}(v^*(c, g), c)$.

- **Maximum of averages** (max-avg): Compute the average cost in each group; return the worst among them; $\mathsf{max\text{-}avg}(c \mid \mathcal{I}) = \max_{g \in \mathcal{G}}(\frac{1}{n_g} \sum_{v \in g} \mathsf{d}(v, c)) = \mathsf{cost}_{g^*(c)}(c)$.

- **Maximum of maxima** (max-max): Return the cost of the most dissatisfied voter overall; $\mathsf{max\text{-}max}(c \mid \mathcal{I}) = \max_{g \in \mathcal{G}} \max_{v \in g} \mathsf{d}(v, c) = \mathsf{d}(v^{**}(c), c)$.

For simplicity, when the objective is clear from context, we simply write $\mathsf{cost}(c \mid \mathcal{I})$. When the instance is also clear, we omit it entirely.

A voting rule f maps a preference profile $\pi$ to a winning candidate w. This rule may be *deterministic* or involve *randomization*. Next, we define a distributed voting mechanism $\Psi = (\mathsf{f}_{in}, \mathsf{f}_{ov})$, which consists of two stages:

- **Stage 1:** Each group $g$ independently selects a *representative* candidate $\mathsf{w}_g$ by applying the in-group rule $\mathsf{f}_{in}$ to the preferences of its members. Let $R = \{\mathsf{w}_g \mid g \in \mathcal{G}\}$.

- **Stage 2:** The final outcome is chosen by applying the over-group rule $\mathsf{f}_{ov}$ to the (centralized) instance $(R, R, \pi^R, \mathsf{d})$, where $R$ acts as *both the set of candidates and the set of voters*, and $\pi^R$ denotes the preferences of representatives over one another.

For each group $g$, the rule $\mathsf{f}_{in}$ has access only to local information: the group size ($n_g$) and the preference profile of its voters ($\pi^g$) over all candidates. In contrast, $\mathsf{f}_{ov}$ receives the preferences of the selected representatives together with the sizes of all groups.

**Distortion.** Given a distributed mechanism $\Psi = (\mathsf{f}_{in}, \mathsf{f}_{ov})$, which takes an instance $\mathcal{I} = (\mathcal{V}, \mathcal{C}, \mathcal{G}, \pi, \mathsf{d})$ as input and outputs a winner $\mathsf{w} = \Psi(\mathcal{I})$, the expected cost of w is defined as $\mathrm{E}[\mathsf{cost}(\mathsf{w} \mid \mathcal{I})] = \sum_{g \in \mathcal{G}} \mathrm{Pr}(\mathsf{w} = \mathsf{w}_g) \cdot \mathrm{E}[\mathsf{cost}(\mathsf{w}_g \mid \mathcal{I})]$, and the expected cost of $\mathsf{w}_g$ is defined as $\mathrm{E}[\mathsf{cost}(\mathsf{w}_g \mid \mathcal{I})] = \sum_{c \in \mathcal{C}} \mathrm{Pr}(\mathsf{w}_g = c) \cdot \mathsf{cost}(c \mid \mathcal{I})$, where $\mathsf{cost}(\cdot)$ denotes one of the four objectives defined earlier. Now, we define the distortion of $\Psi$ as

$$\mathsf{D}(\Psi) = \sup_{\mathcal{I}} \frac{\mathrm{E}[\mathsf{cost}(\Psi(\mathcal{I}) \mid \mathcal{I})]}{\min_{c \in \mathcal{C}} \mathsf{cost}(c \mid \mathcal{I})}.$$

**Definition 2.1** (Pareto Efficiency). *A voting rule is* Pareto efficient *if, for any pair of candidates $x$ and $y$, if all voters prefer $x$ to $y$, then the rule does not select $y$ as the winner.*

We now present several voting rules (for centralized settings) and mechanisms (for distributed settings) frequently used in this work:

- **Plurality Matching rule** ($\mathsf{f}_{pm}$): Introduced by [14], Plurality Matching is a deterministic voting rule that guarantees a distortion of 3, the best possible among all deterministic rules in the metric setting. Given an instance $\mathcal{I}$ with voters $\mathcal{V}$, candidates $\mathcal{C}$, and preference profile $\sigma$, $\mathsf{f}_{pm}$ considers each candidate $c$ and constructs a bipartite domination graph $G(c)$ with voters on both sides and an edge $(v_i, v_j)$ if and only if voter $v_i$ prefers $c$ to top-ranked candidate of voter $v_j$ (i.e., $c \succeq_{v_i} \mathsf{top}(v_j)$). The rule then selects any candidate $c$ for which its corresponding graph $G(c)$ admits a perfect matching (which is guaranteed to exist).

- **Pareto-efficient Plurality Matching rule** ($\mathsf{f}_{pm-par}$): We now introduce a variant of $\mathsf{f}_{pm}$ that ensures Pareto efficiency while preserving the optimal distortion of 3. $\mathsf{f}_{pm-par}$ operates as follows: it starts with a candidate $c_1$ chosen by $\mathsf{f}_{pm}$ (one whose corresponding domination graph $G(c_1)$ admits a perfect matching $M$) and iteratively refines this choice;
    - If $c_1$ is Pareto efficient—no other candidate $c_2$ is preferred by all voters over $c_1$—then $c_1$ is returned as the final winner.
    - Otherwise, there must exist a candidate $c_2$ that Pareto dominates $c_1$ (i.e., every voter prefers $c_2$ to $c_1$). A key insight is that if $c_2$ Pareto dominates $c_1$, then $G(c_2)$ is also admits the same perfect matching $M$ as $G(c_1)$.
    - The voting rule replaces $c_1$ with $c_2$ and iterates until a candidate satisfying Pareto efficiency is found.

- **Random Dictatorship rule** ($\mathsf{f}_{rd}$): A randomized voting rule in which a voter is selected uniformly at random, and the outcome is that voter's top-ranked candidate [15, 37]. In the first stage of the rand-rand mechanism used to establish upper bounds, we apply this rule.

- **Uniform selection rule** ($f_{ur}$): A randomized voting rule that selects the winner uniformly at random from the set of candidates. This rule is used in the second stage of both the rand-det and rand-rand mechanisms that establish upper bounds.

- **Arbitrary Dictator mechanism** ($m_{ad}$): Introduced by [1], this mechanism operates in two stages. First, each group selects a representative by arbitrarily choosing a voter and taking her top-ranked candidate. Second, the final winner is determined by arbitrarily selecting one of the representatives. This mechanism is employed to analyze the upper bound of the max-max objective in det-det.

- $\alpha$-**in-**$\beta$-**over mechanism** ($m_{\alpha\beta}$): Proposed by [1], the $\alpha$-in-$\beta$-over mechanism operates in two deterministic stages, first applying an in-group voting rule with distortion at most $\alpha$, followed by selecting a final winner using an over-group rule with distortion at most $\beta$.

**Definition 2.2** (Promotion). *Given an order $\sigma$ over a set of candidates and a candidate $c \in \sigma$, the operation $\sigma \uparrow c$ returns a new preference $\sigma'$ in which $c$ is moved to the top, and the relative order of all other candidates remains unchanged. When multiple $\uparrow$ operations are applied in sequence, they are evaluated from left to right. $\sigma \uparrow b \uparrow a$ first moves $b$ to the top of $\sigma$, then moves $a$ to the top of $\sigma \uparrow b$.*

Using the promote operation, we define the Bias Tournament—a special complete directed graph (tournament) over the candidates—which is crucial for establishing lower bounds of the rand-det and det-det mechanisms.

**Definition 2.3** (Bias Tournament). *Let $f$ be a deterministic voting rule, $\mathcal{C}$ a set of candidates, and $\sigma$ an ordering of $\mathcal{C}$. The Bias Tournament $\mathcal{T}(f, \mathcal{C}, \sigma)$ is a complete directed graph where each vertex corresponds to a candidate in $\mathcal{C}$. For every pair of distinct candidates $u$ and $w$, there is a directed edge from $u$ to $w$ if and only if $f$ selects $u$ as the winner in a two-voter election with preferences $\pi_1 = \sigma \uparrow w \uparrow u$ and $\pi_2 = \sigma \uparrow u \uparrow w$.*

**Example 1.** *Let $\mathcal{C} = \{c_1, c_2, c_3\}$ and a deterministic rule $f$ that selects the candidate with the smallest index among those ranked first by at least one voter. Suppose: (i) between $c_1$ and $c_2$, the winner is $c_1$; (ii) between $c_1$ and $c_3$, the winner is $c_1$; (iii) between $c_2$ and $c_3$, the winner is $c_2$. Then $\mathcal{T}(f, \mathcal{C}, \sigma)$ contains edges $c_1 \to c_2$, $c_1 \to c_3$, and $c_2 \to c_3$.*

**Basic Observations.** We present several preliminary observations that lay the groundwork for proving our main theorems. For clarity and consistency, we fix an instance $\mathcal{I} = (\mathcal{V}, \mathcal{C}, \mathcal{G}, \pi, d)$ throughout this section. The proof of Observation 2.6 is deferred to the appendix.

**Observation 2.1.** *In distributed voting with single-voter groups, let $\Psi = (f_{in}, f_{ov})$ be a distributed mechanism with finite distortion. Within each group, $f_{in}$ must select the top-ranked candidate of each voter as the group representative.*

**Observation 2.2.** *For any deterministic voting rule $f$ and an ordering $\sigma$ over $\mathcal{C}$, there exists a candidate with in-degree at least $\lceil \frac{m-1}{2} \rceil$ in $\mathcal{T}(f, \mathcal{C}, \sigma)$.*

**Observation 2.3.** *Since $o_g$ is the optimal candidate in group $g$, we have $cost_g(o_g) \leq cost_g(c)$ for any candidate $c$, including $o$. This holds for all objectives considered.*

**Observation 2.4.** *Since $cost(.)$ is defined as the maximum over $cost_g(.)$ under the max-avg and max-max objectives, it follows that $cost_g(o) \leq cost(o)$, for each group $g$.*

**Observation 2.5.** *For rand-det mechanism $\Psi = (f_{in}, f_{ur})$ with output $w$, the expected cost of the mechanism is given by $\mathrm{E}[cost(w)] = \frac{1}{k} \sum_{g \in \mathcal{G}} cost(w_g)$.*

**Observation 2.6.** *For rand-rand mechanism $\Psi = (f_{rd}, f_{ur})$ with output $w$, the expected cost of the mechanism is given by $\mathrm{E}[cost(w)] = \frac{1}{k} \sum_{g \in \mathcal{G}} \frac{1}{n_g} \sum_{v \in g} cost(top(v))$.*

**Observation 2.7.** *For the max-avg objective and any group $g$, we have $cost_g(o) \leq cost(o)$, as implied directly by the definition of max-avg.*

**Observation 2.8.** *Since $top(v)$ denotes the candidate closest to voter $v$, it follows that $d(v, top(v)) \leq d(v, c)$ for any candidate $c$.*

**Observation 2.9.** *For every voter $v$ and every candidate $c$, we have $d(v, c) \leq d(v^{**}(c), c)$.*

**Observation 2.10.** *For every group $g$, every voter $v \in g$, and every candidate $c$, we have $d(v, c) \leq d(v^*(c, g), c)$.*

**Observation 2.11.** *Consider a distributed mechanism $\Psi = (f_{in}, f_{ov})$, where $f_{in}$ is a deterministic rule with distortion at most $\alpha$. By the definition of centralized distortion, for any group $g \in \mathcal{G}$, we obtain $cost_g(w_g) \leq \alpha \cdot cost_g(o_g)$.*

**Observation 2.12.** *Consider a det-det mechanism $\Psi = (f_{in}, f_{ov})$, where $f_{ov}$ has distortion at most $\beta$ with respect to the avg objective. By the definition of centralized distortion, for any group $g \in \mathcal{G}$, we have $\frac{1}{k} \sum_{g \in \mathcal{G}} d(w, w_g) \leq \beta \cdot \frac{1}{k} \sum_{g \in \mathcal{G}} d(o, w_g)$.*

# 3 Distortion Bounds for rand-det

This section examines rand-det mechanisms, defined as pairs $(f_{in}, f_{ov})$, where $f_{in}$ is a deterministic voting rule and $f_{ov}$ is a randomized one. We establish distortion bounds under all considered objectives. For brevity, all proofs are deferred to the appendix.

## 3.1 Upper Bounds

Let $f_\alpha$ be a deterministic voting rule with distortion at most $\alpha$, and let $f_{par}$ be any deterministic voting rule that satisfies Pareto efficiency. We analyze the mechanisms $(f_\alpha, f_{ur})$ and $(f_{par}, f_{ur})$.

For the max-avg and avg-avg objectives, we show that the mechanism $(f_\alpha, f_{ur})$ achieves distortion at most $\alpha + 2$ and $\alpha + 2 - 2/k$, respectively. Since $f_{pm-par}$ achieves the best-known distortion of 3 and also satisfies Pareto efficiency, we instantiate our theorems with the mechanism $(f_{pm-par}, f_{ur})$ to obtain the tightest bounds.

Finally, for the avg-max and max-max objectives, we prove that the mechanism $(f_{par}, f_{ur})$ achieves distortion at most 3, a consequence of Pareto efficiency.

**Theorem 3.1.** *For the max-avg objective in general metric spaces, we have $D((f_\alpha, f_{ur})) \leq \alpha + 2$.*

By using $f_{pm}$ instead of $f_\alpha$ as the in-group voting rule (with $\alpha = 3$) and applying Theorem 3.1, we conclude that $\Psi = (f_{pm}, f_{ur})$ is a rand-det mechanism that satisfies the bound stated in Corollary 3.2.

**Corollary 3.2** (of theorem 3.1). *For the max-avg objective in general metric spaces, there exists a rand-det mechanism with distortion at most 5.*

**Theorem 3.3.** *For the avg-avg objective in general metric spaces, we have $D((f_\alpha, f_{ur})) \leq \alpha + 2 - \frac{2}{k}$.*

Once again, by using $f_{pm}$ instead of $f_\alpha$ as the in-group voting rule (with $\alpha = 3$) and applying Theorem 3.3, we conclude that $\Psi = (f_{pm}, f_{ur})$ is a rand-det mechanism that satisfies the bound stated in Corollary 3.4.

**Corollary 3.4** (of theorem 3.3). *For the avg-avg objective in general metric spaces, there exists a rand-det mechanism with distortion at most $5 - 2/k$.*

For the avg-max and max-max objectives, we derive an upper bound of 5 following an argument analogous to the proof of Theorem 3.1. Now, we improve the upper bound of 5 to 3 by applying the property of Pareto efficiency.

**Theorem 3.5.** *For the avg-max and max-max objectives in general metric spaces, we have $D((f_{par}, f_{ur})) \leq 3$.*

## 3.2 Lower Bounds

Now, we establish lower bounds on the distortion of rand-det mechanisms. Specifically, Theorem 3.6 provides lower bounds for the max-max and avg-max objectives, Theorem 3.7 covers the max-avg objective, and Theorem 3.8 addresses the avg-avg objective. It is worth noting that all the lower bounds in this section are derived from symmetric instances and apply in that setting as well.

**Theorem 3.6.** *For the avg-max and max-max objectives, the distortion of any rand-det mechanism is at least 3, even when the metric space is a line.*

For the max-avg, and avg-avg, objectives, we use the Bias Tournament to establish the lower bounds stated in Theorems 3.7 and 3.8.

**Theorem 3.7.** *For the max-avg objective, the distortion of any rand-det mechanism is at least 5, even when the metric space is a line.*

We now establish the lower bound for the avg-avg objective in general metric spaces.

**Theorem 3.8.** *For general metric spaces and the avg-avg objective, the distortion of any rand-det mechanism is at least $5 - \frac{2}{k}$.*

# 4 Distortion Bounds for rand-rand

This section examines the rand-rand mechanisms, which are pairs $(f_{in}, f_{ov})$ composed of two randomized voting rules $f_{in}$ and $f_{ov}$. We establish distortion bounds under all considered objectives.

## 4.1 Upper Bounds

Throughout this section, we analyze the mechanism $(f_{rd}, f_{ur})$ and show that despite its simplicity, it achieves tight (or nearly tight) distortion bounds for various cost objectives. In particular, for the max-max objective (indeed the max objective), we establish that choosing the top candidate of any voter yields a distortion of at most 3. We begin with the simplest case, max-max, and move towards the most intricate avg-avg.

**Theorem 4.1.** *For the max-max objective in general metric spaces, we have* $D((f_{rd}, f_{ur})) \leq 3$.

**Theorem 4.2.** *For the avg-max objective in general metric spaces, we have* $D((f_{rd}, f_{ur})) \leq 3$.

For the max-avg objective, the key insight is to show that for any voter $v$, $\mathsf{cost}_{g^*(\mathsf{top}(v))}(\mathsf{top}(v)) \leq 2\mathsf{d}(v, \mathsf{o}) + \mathsf{cost}(\mathsf{o})$. This crucial inequality is the foundation for proving the desired upper bound.

**Theorem 4.3.** *For the max-avg objective in general metric spaces, we have* $D((f_{rd}, f_{ur})) \leq 3$.

**Theorem 4.4.** *For the avg-avg objective in general metric spaces, we have* $D((f_{rd}, f_{ur})) \leq 3 - 2/kn^*$ *where $n^*$ represents the maximum value of $n_g$ over all groups.*

As a corollary of Theorem 4.4, we conclude that $\Psi = (f_{rd}, f_{ur})$ is a rand-rand mechanism that satisfies the bound stated in Corollary 4.5, particularly in the symmetric case $(kn^* = n)$.

**Corollary 4.5** (of theorem 4.4). *For the avg-avg objective in general metric spaces, there exists a rand-rand mechanism with distortion at most $3 - \frac{2}{n}$, provided that the groups are symmetric.*

## 4.2 Lower Bounds

In this section, all results are derived from the symmetric instances and apply to that setting as well.

**Theorem 4.6.** *For the max-avg and max-max objectives, the distortion of any rand-rand mechanism is at least 3, even when the metric space is a line.*

**Theorem 4.7.** *For the avg-max objective, the distortion of any rand-rand mechanism is at least $3 - \frac{2}{n}$ (equivalently, $3 - \frac{2}{m}$), even when the metric space is a line.*

In the centralized setting, where all voters belong to a single group $(k = 1)$, the avg-max objective simplifies to the max objective. Thus, the instance and analysis from the proof of Theorem 4.7, which already considers the single-group case, apply directly. As the number of voters $n$ increases, the distortion approaches 3. Therefore, for any constant $\varepsilon > 0$, an instance can be constructed with a sufficiently large number of voters $(n > \frac{2}{\varepsilon})$ such that the distortion of any randomized voting rule exceeds $3 - \varepsilon$. This yields the following corollary.

**Corollary 4.8** (of Theorem 4.7). *For the max objective in the centralized setting, the distortion of any randomized voting rule is at least $3 - \varepsilon$, for any constant $\varepsilon > 0$, even when the metric is a line.*

**Theorem 4.9.** *For general metric spaces and the avg-avg objective, the distortion of any rand-rand mechanism is at least $3 - \frac{2}{n}$ (equivalently $3 - \frac{2}{k}$).*

# 5 Resolving the Distortion Bounds for det-det

In this section, we study the distortion of deterministic distributed mechanisms, providing both lower and upper bounds in general metric spaces. Anshelevich *et al.* [1] proposed the $\alpha$-in-$\beta$-over mechanism $(m_{\alpha\beta})$, which allows any candidate—not just representatives—to be selected as the final winner. We adopt their approach here.

## 5.1 Upper Bounds

We establish improved upper bounds for deterministic mechanisms with respect to the avg-max and max-max objectives. Let $f_\beta$ be a deterministic voting rule with distortion at most $\beta$, and let $f_{par}$

be a deterministic voting rule that satisfies the property of Pareto efficiency. We improve the best known upper bound for the avg-max objective from 11, as proved by [1], to 7 by applying mechanism $(f_{par}, f_\beta)$. For the max-max objective, we improve the previous upper bound of 5 to 3 using $m_{ad}$. We begin with the simplest case, max-max, and move towards the slightly more intricate avg-max.

**Theorem 5.1.** *For the max-max objective in general metric spaces, we have* $D(m_{ad}) \le 3$.

**Theorem 5.2.** *For the avg-max objective in general metric spaces, we have* $D((f_{par}, f_\beta)) \le 2\beta + 3$.

We can apply $f_{pm-par}$ as both the in-group and over-group voting rules. Assuming each voter is at a distance of 0 from her top choice, this yields a distortion of $\beta = 2$, as shown in [14]. Combined with Theorem 5.2, we conclude that $\Psi = (f_{pm-par}, f_{pm-par})$ is a det-det mechanism that satisfies the bound in Corollary 5.3.

**Corollary 5.3** (of Theorem 5.2)**.** *For the avg-max objective in general metric spaces, there exists a det-det mechanism with distortion at most 7.*

### 5.2 Lower Bounds

In this section, we present a lower bound of 5 on the distortion of any det-det mechanisms under the max-avg objective, improving upon the previous bound of $2 + \sqrt{5}$, which achieved by [1]. Following the framework in that paper, our analysis applies the over-group voting rule across the set of candidates ($\mathcal{C}$), rather than the group representatives ($R$).

**Theorem 5.4.** *For general metric spaces and the max-avg objective, the distortion of any det-det mechanism is at least* $5$.

## 6 An Extension of Lower Bounds for rand-rand and rand-det

In this section, we focus on the Euclidean metric and establish lower bounds on the avg-avg distortion for both the rand-rand and rand-det mechanisms, as presented in Theorems 6.1 and 6.2, respectively.

**Theorem 6.1.** *For the avg-avg objective in Euclidean space, the distortion of any rand-rand mechanism is at least* $\sqrt{5} - \varepsilon$, *for every constant* $\varepsilon > 0$.

**Theorem 6.2.** *For the avg-avg objective in Euclidean space, the distortion of any rand-det mechanism is at least* $2 + \sqrt{5} - \varepsilon$, *for every constant* $\varepsilon > 0$.

## 7 Discussion and Open Problems

In this paper, we have initiated the study of metric distortion in single-winner distributed voting under randomized mechanisms (rand-rand and rand-det) for many different objectives. We also have improved upon previous results for deterministic mechanisms (det-det). Although our work presents an almost complete picture in the distortion of distributed voting problem, it reveals several promising directions for future research. A significant challenge about our work leaves open lies in analyzing the det-rand mechanisms, where random decisions in the first stage are followed by deterministic ones in the second. Our primary tool, the Bias Tournament technique, is incompatible with the randomized first stage of det-rand. Currently, our understanding is confined to basic results inherited from rand-rand (for the lower bounds) and det-det (for the upper bounds). To obtain tighter results, we need more refined arguments, a promising direction for future work.

Within the scope of our work, another possible direction could be to close the remaining narrow gaps between the lower and upper bounds presented in Table 2, particularly for the avg-avg and avg-max objectives in det-det, as well as the avg-avg and avg-max objective in rand-rand. Another direction is to examine the distortion of the avg-avg objective for all proposed mechanisms, in more structured spaces—Euclidean and line metrics—where results may differ from those in general metric spaces. Another natural extension is to investigate distributed mechanisms in a cardinal setting, where agents have access to exact distances, instead of solely the ordinal rankings induced by those distances.

Going beyond the single-winner voting, one could study the distortion of distributed mechanisms that select committees comprising a specified number of alternatives. Another intriguing direction for future research is to investigate how agents' strategic behavior impacts distributed distortion. The goal could be to understand whether it is possible to design distributed mechanisms that are both strategyproof and capable of achieving low distortion.

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

# A  Proofs for Section 2 (Basic Notations)

**Observation 2.6.** *For rand-rand mechanism $\Psi = (\mathsf{f}_{rd}, \mathsf{f}_{ur})$ with output $\mathsf{w}$, the expected cost of the mechanism is given by* $\mathrm{E}[\mathsf{cost}(\mathsf{w})] = \frac{1}{k} \sum_{g \in \mathcal{G}} \frac{1}{n_g} \sum_{v \in g} \mathsf{cost}(\mathsf{top}(v))$.

*Proof.* By the definitions of the *Random Dictatorship* rule ($\mathsf{f}_{rd}$) and the uniform selection rule ($\mathsf{f}_{ur}$), we have

$$\mathrm{E}[\mathsf{cost}(\mathsf{w})] = \sum_{g \in \mathcal{G}} \Pr(\mathsf{w} = \mathsf{w}_g) \cdot \mathrm{E}[\mathsf{cost}(\mathsf{w}_g)]$$

$$= \frac{1}{k} \sum_{g \in \mathcal{G}} \mathrm{E}[\mathsf{cost}(\mathsf{w}_g)]$$

$$= \frac{1}{k} \sum_{g \in \mathcal{G}} \sum_{v \in g} \Pr(\mathsf{w}_g = \mathsf{top}(v)) \cdot \mathsf{cost}(\mathsf{top}(v))$$

$$= \frac{1}{k} \sum_{g \in \mathcal{G}} \frac{1}{n_g} \sum_{v \in g} \mathsf{cost}(\mathsf{top}(v)).$$

$\square$

# B  Proofs for Section 3 (Distortion Bounds for rand-det)

**Theorem 3.1.** *For the max-avg objective in general metric spaces, we have* $\mathrm{D}((\mathsf{f}_\alpha, \mathsf{f}_{ur})) \leq \alpha + 2$.

*Proof.* Consider an instance $\mathcal{I} = (\mathcal{V}, \mathcal{C}, \mathcal{G}, \pi, \mathsf{d})$ and rand-det mechanism $\Psi = (\mathsf{f}_\alpha, \mathsf{f}_{ur})$. We have

$$\mathrm{E}[\mathsf{cost}(\mathsf{w})] = \sum_{g \in \mathcal{G}} \frac{1}{k} \cdot \mathsf{cost}(\mathsf{w}_g) \qquad \text{(Observation 2.5)}$$

$$= \sum_{g \in \mathcal{G}} \frac{1}{k} \cdot \frac{1}{n_{g^*(\mathsf{w}_g)}} \sum_{v \in g^*(\mathsf{w}_g)} \mathsf{d}(v, \mathsf{w}_g) \qquad \text{(Definition of max-avg)}$$

$$\leq \sum_{g \in \mathcal{G}} \frac{1}{k} \cdot \frac{1}{n_{g^*(\mathsf{w}_g)}} \sum_{v \in g^*(\mathsf{w}_g)} \left( \mathsf{d}(v, \mathsf{o}) + \mathsf{d}(\mathsf{o}, \mathsf{w}_g) \right) \qquad \text{(Triangle Inequality)}$$

$$= \sum_{g \in \mathcal{G}} \frac{1}{k} \cdot \frac{1}{n_{g^*(\mathsf{w}_g)}} \sum_{v \in g^*(\mathsf{w}_g)} \mathsf{d}(v, \mathsf{o})$$

$$+ \sum_{g \in \mathcal{G}} \frac{1}{k} \cdot \frac{1}{n_{g^*(\mathsf{w}_g)}} \sum_{v \in g^*(\mathsf{w}_g)} \mathsf{d}(\mathsf{o}, \mathsf{w}_g)$$

$$\leq \sum_{g \in \mathcal{G}} \frac{1}{k} \cdot \mathsf{cost}(\mathsf{o}) + \sum_{g \in \mathcal{G}} \frac{1}{k} \cdot \frac{1}{n_{g^*(\mathsf{w}_g)}} \sum_{v \in g^*(\mathsf{w}_g)} \mathsf{d}(\mathsf{o}, \mathsf{w}_g) \qquad \text{(Observation 2.4)}$$

$$= \sum_{g \in \mathcal{G}} \frac{1}{k} \cdot \mathsf{cost}(\mathsf{o}) + \sum_{g \in \mathcal{G}} \frac{1}{k} \cdot \mathsf{d}(\mathsf{o}, \mathsf{w}_g)$$

$$\leq \mathsf{cost}(\mathsf{o}) + \sum_{g \in \mathcal{G}} \frac{1}{k} \cdot \frac{1}{n_g} \sum_{v \in g} \left( \mathsf{d}(v, \mathsf{o}) + \mathsf{d}(v, \mathsf{w}_g) \right) \qquad \text{(Triangle Inequality)}$$

$$= \mathsf{cost}(\mathsf{o}) + \sum_{g \in \mathcal{G}} \frac{1}{k} \cdot \frac{1}{n_g} \sum_{v \in g} \mathsf{d}(v, \mathsf{o})$$

$$+ \sum_{g \in \mathcal{G}} \frac{1}{k} \cdot \frac{1}{n_g} \sum_{v \in g} \mathsf{d}(v, \mathsf{w}_g)$$

$$\leq \mathsf{cost}(\mathsf{o}) + \sum_{g \in \mathcal{G}} \frac{1}{k} \cdot \mathsf{cost}(\mathsf{o}) + \sum_{g \in \mathcal{G}} \frac{1}{k} \cdot \mathsf{cost}_g(\mathsf{w}_g)$$

$$= 2\text{cost}(\text{o}) + \sum_{g \in \mathcal{G}} \frac{1}{k} \cdot \text{cost}_g(\text{w}_g)$$

$$\leq 2\text{cost}(\text{o}) + \sum_{g \in \mathcal{G}} \frac{1}{k} \cdot \alpha \cdot \text{cost}_g(\text{o}_g) \qquad \text{(Observation 2.11)}$$

$$\leq 2\text{cost}(\text{o}) + \alpha \sum_{g \in \mathcal{G}} \frac{1}{k} \cdot \text{cost}_g(\text{o}) \qquad \text{(Observation 2.3)}$$

$$\leq 2\text{cost}(\text{o}) + \alpha \sum_{g \in \mathcal{G}} \frac{1}{k} \cdot \text{cost}(\text{o}) \qquad \text{(Observation 2.4)}$$

$$= (\alpha + 2)\text{cost}(\text{o}).$$

$$\square$$

**Theorem 3.3.** *For the avg-avg objective in general metric spaces, we have* $\text{D}((f_\alpha, f_{ur})) \leq \alpha + 2 - \frac{2}{k}$.

*Proof.* Consider an instance $\mathcal{I} = (\mathcal{V}, \mathcal{C}, \mathcal{G}, \pi, \text{d})$ and rand-det mechanism $\Psi = (f_\alpha, f_{ur})$. We have

$$\mathbb{E}[\text{cost}(\text{w})] = \sum_{g \in \mathcal{G}} \frac{1}{k} \cdot \text{cost}(\text{w}_g) \qquad \text{(Observation 2.5)}$$

$$= \frac{1}{k} \sum_{g \in \mathcal{G}} \frac{1}{k} \sum_{g' \in \mathcal{G}} \text{cost}_{g'}(\text{w}_g)$$

$$= \frac{1}{k} \sum_{g \in \mathcal{G}} \frac{1}{k} \sum_{g' \in \mathcal{G}} \frac{1}{n_{g'}} \sum_{v \in g'} \text{d}(v, \text{w}_g) \qquad \text{(Definition of avg-avg)}$$

$$\leq \frac{1}{k} \sum_{g \in \mathcal{G}} \frac{1}{k} \sum_{g' \in \mathcal{G}, g \neq g'} \frac{1}{n_{g'}} \sum_{v \in g'} \Big(\text{d}(v, \text{o}) + \text{d}(\text{o}, \text{w}_g)\Big)$$

$$\quad + \frac{1}{k} \sum_{g \in \mathcal{G}} \frac{1}{k} \sum_{g' \in \mathcal{G}, g = g'} \frac{1}{n_{g'}} \sum_{v \in g'} \text{d}(v, \text{w}_g) \qquad \text{(Triangle Inequality)}$$

$$= \frac{1}{k} \sum_{g \in \mathcal{G}} \frac{1}{k} \sum_{g' \in \mathcal{G}, g \neq g'} \frac{1}{n_{g'}} \sum_{v \in g'} \Big(\text{d}(v, \text{o}) + \text{d}(\text{o}, \text{w}_g)\Big)$$

$$\quad + \frac{1}{k} \sum_{g \in \mathcal{G}} \frac{1}{k} \cdot \text{cost}_g(\text{w}_g)$$

$$\leq \frac{1}{k} \sum_{g \in \mathcal{G}} \frac{1}{k} \sum_{g' \in \mathcal{G}, g \neq g'} \frac{1}{n_{g'}} \sum_{v \in g'} \Big(\text{d}(v, \text{o}) + \text{d}(\text{o}, \text{w}_g)\Big)$$

$$\quad + \frac{1}{k} \sum_{g \in \mathcal{G}} \frac{1}{k} \cdot \alpha \cdot \text{cost}_g(\text{o}_g) \qquad \text{(Observation 2.11)}$$

$$\leq \frac{1}{k} \sum_{g \in \mathcal{G}} \frac{1}{k} \sum_{g' \in \mathcal{G}, g \neq g'} \frac{1}{n_{g'}} \sum_{v \in g'} \Big(\text{d}(v, \text{o}) + \text{d}(\text{o}, \text{w}_g)\Big)$$

$$\quad + \frac{1}{k} \sum_{g \in \mathcal{G}} \frac{1}{k} \cdot \alpha \cdot \text{cost}_g(\text{o}) \qquad \text{(Observation 2.3)}$$

$$= \frac{1}{k} \sum_{g \in \mathcal{G}} \frac{1}{k} \sum_{g' \in \mathcal{G}, g \neq g'} \frac{1}{n_{g'}} \sum_{v \in g'} \Big(\text{d}(v, \text{o}) + \text{d}(\text{o}, \text{w}_g)\Big)$$

$$\quad + \frac{\alpha}{k} \cdot \text{cost}(\text{o}) \qquad \Big(\text{cost}(\text{o}) = \sum_{g \in \mathcal{G}} \frac{1}{k}\text{cost}_g(\text{o})\Big)$$

$$= \frac{1}{k} \sum_{g \in \mathcal{G}} \frac{1}{k} \sum_{g' \in \mathcal{G}, g \neq g'} \frac{1}{n_{g'}} \sum_{v \in g'} \text{d}(v, \text{o})$$

$$+ \frac{1}{k} \sum_{g \in \mathcal{G}} \frac{1}{k} \sum_{g' \in \mathcal{G}, g \neq g'} \frac{1}{n_{g'}} \sum_{v \in g'} \mathsf{d}(\mathsf{o}, \mathsf{w}_g) + \frac{\alpha}{k} \cdot \mathsf{cost}(\mathsf{o})$$

$$= \frac{1}{k} \sum_{g \in \mathcal{G}} \frac{1}{k} \sum_{g' \in \mathcal{G}, g \neq g'} \frac{1}{n_{g'}} \sum_{v \in g'} \mathsf{d}(v, \mathsf{o})$$

$$+ \frac{1}{k} \sum_{g \in \mathcal{G}} \frac{k-1}{k} \cdot \mathsf{d}(\mathsf{o}, \mathsf{w}_g) + \frac{\alpha}{k} \cdot \mathsf{cost}(\mathsf{o})$$

$$\leq \frac{1}{k} \sum_{g \in \mathcal{G}} \frac{1}{k} \sum_{g' \in \mathcal{G}, g \neq g'} \frac{1}{n_{g'}} \sum_{v \in g'} \mathsf{d}(v, \mathsf{o})$$

$$+ \frac{1}{k} \sum_{g \in \mathcal{G}} \frac{k-1}{k} \cdot \frac{1}{n_g} \sum_{v \in g} (\mathsf{d}(v, \mathsf{o}) + \mathsf{d}(v, \mathsf{w}_g)) + \frac{\alpha}{k} \cdot \mathsf{cost}(\mathsf{o}) \qquad \text{(Triangle Inequality)}$$

$$= \frac{1}{k} \sum_{g \in \mathcal{G}} \frac{1}{k} \sum_{g' \in \mathcal{G}, g \neq g'} \frac{1}{n_{g'}} \sum_{v \in g'} \mathsf{d}(v, \mathsf{o})$$

$$+ \frac{1}{k} \sum_{g \in \mathcal{G}} \frac{k-1}{k} \cdot (\mathsf{cost}_g(\mathsf{o}) + \mathsf{cost}_g(\mathsf{w}_g)) + \frac{\alpha}{k} \cdot \mathsf{cost}(\mathsf{o}) \qquad \text{(Definition of } \mathsf{cost}_g(.))$$

$$\leq \frac{1}{k} \sum_{g \in \mathcal{G}} \frac{1}{k} \sum_{g' \in \mathcal{G}, g \neq g'} \frac{1}{n_{g'}} \sum_{v \in g'} \mathsf{d}(v, \mathsf{o})$$

$$+ \frac{1}{k} \sum_{g \in \mathcal{G}} \frac{k-1}{k} \cdot (\mathsf{cost}_g(\mathsf{o}) + \alpha \cdot \mathsf{cost}_g(\mathsf{o}_g)) + \frac{\alpha}{k} \cdot \mathsf{cost}(\mathsf{o}) \qquad \text{(Observation 2.11)}$$

$$\leq \frac{1}{k} \sum_{g \in \mathcal{G}} \frac{1}{k} \sum_{g' \in \mathcal{G}, g \neq g'} \frac{1}{n_{g'}} \sum_{v \in g'} \mathsf{d}(v, \mathsf{o})$$

$$+ \frac{1}{k} \sum_{g \in \mathcal{G}} \frac{k-1}{k} \cdot (\mathsf{cost}_g(\mathsf{o}) + \alpha \cdot \mathsf{cost}_g(\mathsf{o})) + \frac{\alpha}{k} \cdot \mathsf{cost}(\mathsf{o}) \qquad \text{(Observation 2.3)}$$

$$= \frac{1}{k} \sum_{g \in \mathcal{G}} \frac{1}{k} \sum_{g' \in \mathcal{G}, g \neq g'} \frac{1}{n_{g'}} \sum_{v \in g'} \mathsf{d}(v, \mathsf{o})$$

$$+ \frac{(k-1)(\alpha+1)}{k} \cdot \mathsf{cost}(\mathsf{o}) + \frac{\alpha}{k} \cdot \mathsf{cost}(\mathsf{o}) \qquad \text{(Definition of avg-avg)}$$

$$= \frac{1}{k} \sum_{g \in \mathcal{G}} \frac{1}{k} \sum_{g' \in \mathcal{G}, g \neq g'} \mathsf{cost}_{g'}(\mathsf{o}) + \frac{\alpha k + k - 1}{k} \cdot \mathsf{cost}(\mathsf{o})$$

$$= \frac{1}{k} \sum_{g \in \mathcal{G}} \frac{1}{k} \sum_{g' \in \mathcal{G}} \mathsf{cost}_{g'}(\mathsf{o}) - \frac{1}{k} \sum_{g \in \mathcal{G}} \frac{1}{k} \sum_{g' \in \mathcal{G}, g = g'} \mathsf{cost}_{g'}(\mathsf{o})$$

$$+ \frac{\alpha k + k - 1}{k} \cdot \mathsf{cost}(\mathsf{o})$$

$$= \mathsf{cost}(\mathsf{o}) - \frac{1}{k} \sum_{g \in \mathcal{G}} \frac{1}{k} \cdot \mathsf{cost}_g(\mathsf{o}) + \frac{\alpha k + k - 1}{k} \cdot \mathsf{cost}(\mathsf{o})$$

$$= \mathsf{cost}(\mathsf{o}) - \frac{1}{k} \cdot \mathsf{cost}(\mathsf{o}) + \frac{\alpha k + k - 1}{k} \cdot \mathsf{cost}(\mathsf{o})$$

$$= (\alpha + 2 - \frac{2}{k}) \mathsf{cost}(\mathsf{o}).$$

□

**Theorem 3.5.** *For the avg-max and max-max objectives in general metric spaces, we have* $\mathsf{D}((\mathsf{f}_{par}, \mathsf{f}_{ur})) \leq 3$.

*Proof.* We present a proof for the avg-max objective. A similar argument can be used to prove the result for the max-max objective as well.

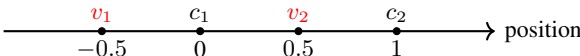

Figure 1: An example used in the proof of Theorem 3.6.

Consider an instance $\mathcal{I} = (\mathcal{V}, \mathcal{C}, \mathcal{G}, \pi, \mathsf{d})$ and rand-det mechanism $\Psi = (\mathsf{f}_{par}, \mathsf{f}_{ur})$. By the property of Pareto efficiency, for each group $g$, there exists a voter $v_g \in g$ who prefers $\mathsf{w}_g$ to $\mathsf{o}$. Therefore, we have

$$\mathrm{E}[\mathsf{cost}(\mathsf{w})] = \sum_{g \in \mathcal{G}} \frac{1}{k} \cdot \mathsf{cost}(\mathsf{w}_g) \qquad\qquad \text{(Observation 2.5)}$$

$$= \sum_{g \in \mathcal{G}} \frac{1}{k} \cdot \frac{1}{k} \sum_{g' \in \mathcal{G}} \mathsf{d}(v^*(\mathsf{w}_g, g'), \mathsf{w}_g) \qquad\qquad \text{(Definition of avg-max)}$$

$$\leq \sum_{g \in \mathcal{G}} \frac{1}{k} \cdot \frac{1}{k} \sum_{g' \in \mathcal{G}} \left( \mathsf{d}(v^*(\mathsf{w}_g, g'), \mathsf{o}) + \mathsf{d}(\mathsf{o}, \mathsf{w}_g) \right) \qquad\qquad \text{(Triangle Inequality)}$$

$$\leq \sum_{g \in \mathcal{G}} \frac{1}{k} \cdot \frac{1}{k} \sum_{g' \in \mathcal{G}} \left( \mathsf{d}(v^*(\mathsf{o}, g'), \mathsf{o}) + \mathsf{d}(\mathsf{o}, \mathsf{w}_g) \right) \qquad\qquad \text{(Observation 2.10)}$$

$$= \mathsf{cost}(\mathsf{o}) + \sum_{g \in \mathcal{G}} \frac{1}{k} \cdot \mathsf{d}(\mathsf{o}, \mathsf{w}_g)$$

$$\leq \mathsf{cost}(\mathsf{o}) + \frac{1}{k} \sum_{g \in \mathcal{G}} \left( \mathsf{d}(\mathsf{o}, v_g) + \mathsf{d}(v_g, \mathsf{w}_g) \right) \qquad\qquad \text{(Triangle Inequality)}$$

$$\leq \mathsf{cost}(\mathsf{o}) + \frac{2}{k} \sum_{g \in \mathcal{G}} \mathsf{d}(v_g, \mathsf{o}) \qquad\qquad (\mathsf{d}(v_g, \mathsf{w}_g) \leq \mathsf{d}(v_g, \mathsf{o}))$$

$$\leq \mathsf{cost}(\mathsf{o}) + \frac{2}{k} \sum_{g \in \mathcal{G}} \mathsf{d}(v^*(\mathsf{o}, g), \mathsf{o}) \qquad\qquad \text{(Observation 2.10)}$$

$$= 3\mathsf{cost}(\mathsf{o}).$$

$\square$

**Theorem 3.6.** *For the avg-max and max-max objectives, the distortion of any rand-det mechanism is at least $3$, even when the metric space is a line.*

*Proof.* Consider a rand-det mechanism $\Psi = (\mathsf{f}_{in}, \mathsf{f}_{ov})$. We construct an instance with candidates $\mathcal{C} = \{c_1, c_2\}$ and voters $\mathcal{V} = \{v_1, v_2\}$ in a single group. $c_1$ and $c_2$ are located at positions $0$ and $1$, respectively. $v_1$ and $v_2$ with preference profiles $\pi_1 = (c_1, c_2)$ and $\pi_2 = (c_2, c_1)$, are also positioned at $-0.5$ and $0.5$, respectively. Refer to Figure 1 for a visual illustration. Without loss of generality, assume that $\Psi$ selects $c_2$ as the representative of the group, and thus the final winner is $c_2$. Since there is only one group, the avg-max and max-max objectives both simplify to max. Thus, we have $\mathsf{cost}(c_1) = \frac{1}{2}$ and $\mathsf{cost}(c_2) = \frac{3}{2}$. Clearly, $c_1$ is the optimal candidate. The distortion of $\Psi$ is

$$\mathsf{D}(\Psi) \geq \frac{\mathsf{cost}(c_2)}{\mathsf{cost}(c_1)} = 3.$$

$\square$

**Theorem 3.7.** *For the max-avg objective, the distortion of any rand-det mechanism is at least $5$, even when the metric space is a line.*

*Proof.* Consider a rand-det mechanism $\Psi = (\mathsf{f}_{in}, \mathsf{f}_{ov})$. We construct an instance with candidates $\mathcal{C} = \{c_1, c_2, c_3, c_4\}$, and voters $\mathcal{V} = \{v_1, v_2, v_3, v_4\}$, all located along a line metric. The voters are partitioned into two groups, $g_1 = \{v_1, v_2\}$ and $g_2 = \{v_3, v_4\}$. Let $\sigma$ be an arbitrary ordering of the candidates. Without loss of generality, assume that $c_1$ is a candidate with in-degree at least

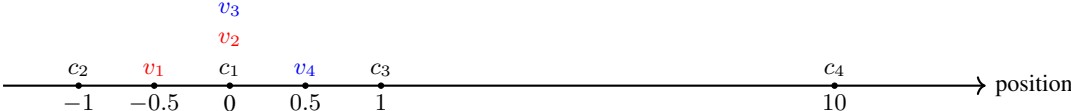

Figure 2: An example used in case 1 of Theorem 3.7. Different voter groups are distinguished by distinct colors.

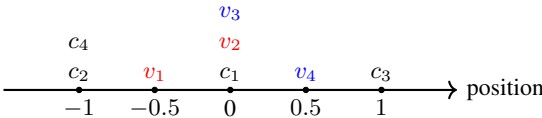

Figure 3: An example used in case 2 of Theorem 3.7. Different voter groups are distinguished by distinct colors.

$\left\lceil \frac{m-1}{2} \right\rceil = 2$ in tournament $\mathcal{T}(\mathsf{f}_{in}, \mathcal{C}, \sigma)$, such a candidate is guaranteed to exist by Observation 2.2. Suppose $c_2$ and $c_3$ are two candidates with directed edges toward $c_1$, meaning that $c_1$ is the "losing" candidate while both $c_2$ and $c_3$ "defeat" it in the tournament. We may further assume that $c_2 \succ_\sigma c_3$. Now, consider the following construction on the line metric:

- Voters $v_2$ and $v_3$ are located at positions $0$, while voters $v_1$ and $v_4$ are located at $-0.5$ and $0.5$, respectively.

- Candidates $c_2$, $c_1$, and $c_3$ are located at positions $-1$, $0$, and $1$, respectively. The position of candidate $c_4$ depends on the ordering $\sigma$, ensuring the input to $\mathcal{T}(\mathsf{f}_{in}, \mathcal{C}, \sigma)$ remains valid. We analyze three cases based on the relative ordering of $c_2, c_3, c_4$:

  - **Case 1:** If $c_2 \succ_\sigma c_3 \succ_\sigma c_4$, candidate $c_4$ is located at position $10$. Refer to Figure 2 for an illustration.
  - **Case 2:** If $c_2 \succ_\sigma c_4 \succ_\sigma c_3$, candidate $c_4$ is located at position $-1$. This case is illustrated in Figure 3
  - **Case 3:** If $c_4 \succ_\sigma c_2 \succ_\sigma c_3$, candidate $c_4$ is located at $1$. A visual representation of this case can be found in Figure 4

Note that when a voter is equidistant from two candidates, multiple preference profiles may be consistent with the underlying metric space. According to $\mathcal{T}(\mathsf{f}_{in}, \mathcal{C}, \sigma)$, we can determine the group representatives:

- A directed edge from $c_2$ to $c_1$, implies that $c_2$ is selected as the representative for group $g_1$.

- Similarly, a directed edge from $c_3$ to $c_1$, means $c_3$ is the representative for group $g_2$.

By the definition of the max-avg objective, we have $\mathsf{cost}(c_2) = \mathsf{cost}(c_3) = \frac{5}{4}$ and $\mathsf{cost}(c_1) = \frac{1}{4}$. Thus, $c_1$ is the optimal candidate in all cases. The mechanism must select the final winner from the group representatives, $c_2$ or $c_3$. Finally, we derive the distortion of mechanism $\Psi$ as:

$$\mathsf{D}(\Psi) \geq \min\left( \frac{\mathsf{cost}(c_2)}{\mathsf{cost}(o)}, \frac{\mathsf{cost}(c_3)}{\mathsf{cost}(o)} \right)$$
$$= \frac{\frac{5}{4}}{\mathsf{cost}(c_1)}$$
$$= 5.$$

$\square$

**Theorem 3.8.** *For general metric spaces and the avg-avg objective, the distortion of any rand-det mechanism is at least $5 - \frac{2}{k}$.*

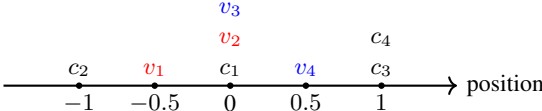

Figure 4: An example used in case 3 of Theorem 3.7. Different voter groups are distinguished by distinct colors.

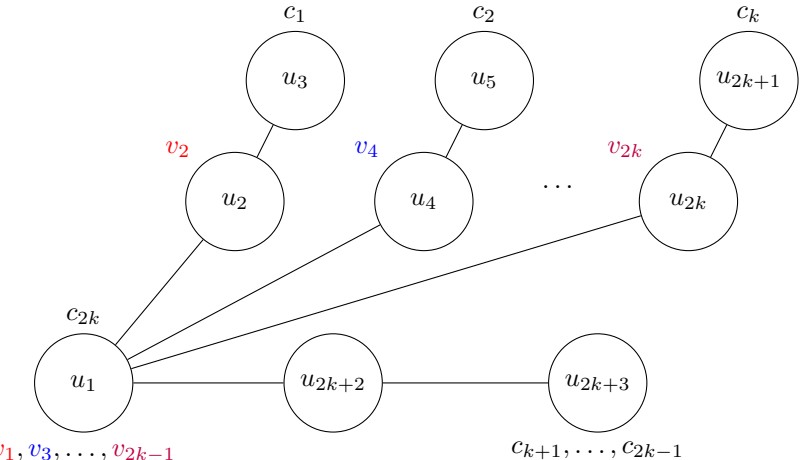

Figure 5: Tree graph used in the proof of Theorem 3.8. Different voter groups are distinguished by distinct colors.

*Proof.* Consider a rand-det mechanism $\Psi = (f_{in}, f_{ov})$. We construct an instance with a set of $m = 2k$ candidates, $\mathcal{C} = \{c_1, c_2, \ldots, c_{m=2k}\}$, a set of $n = 2k$ voters, $\mathcal{V} = \{v_1, v_2, \ldots, v_{n=2k}\}$, and $k$ groups $g_i = \{v_{2i-1}, v_{2i}\}$ for $1 \leq i \leq k$. Let $\sigma$ be an arbitrary ordering of the candidates. Without loss of generality, assume that $c_{2k}$ is a candidate with in-degree at least $\lceil \frac{m-1}{2} \rceil = k$ in the tournament $\mathcal{T}(f_{in}, \mathcal{C}, \sigma)$, such a candidate is guaranteed to exist by Observation 2.2. Further, suppose $c_1, c_2, \ldots, c_k$ are $k$ candidates that have directed edges toward $c_{2k}$ in the tournament, meaning that $c_{2k}$ is the "losing" candidate.

We construct a connected graph $G$ with $2k + 3$ vertices, denoted $u_1, u_2, \ldots, u_{2k+3}$, where the shortest-path distances in $G$ define the underlying metric space d. Each voter and candidate is placed on one of the vertices (a single vertex may host multiple entities). The graph $G$ is constructed as follows (see Figure 5 for an illustration):

- Place candidate $c_{2k}$ at vertex $u_1$.

- For each $1 \leq i \leq k + 1$, add an edge between $u_1$ and $u_{2i}$, and another edge between $u_{2i}$ and $u_{2i+1}$. This forms $k + 1$ branches extending from the central vertex $u_1$.

- For each $1 \leq i \leq k$, place voter $v_{2i-1}$ at vertex $u_1$, voter $v_{2i}$ at vertex $u_{2i}$, and candidate $c_i$ at vertex $u_{2i+1}$.

- For each $k + 1 \leq i \leq 2k - 1$, place candidate $c_i$ at vertex $u_{2k+3}$.

Pairwise distances between the candidates and voters are presented in Tables 3 and 4. Moreover, the preference profiles in Table 5 induced by the shortest-path distances in graph $G$, are consistent with the metric space d. Note that multiple preference profiles may be consistent with d.

According to $\mathcal{T}(f_{in}, \mathcal{C}, \sigma)$, the representative of group $g_i$ is candidate $c_i$ for any $1 \leq i \leq k$. Thus, the mechanism must select one of these representatives as the final winner. By the definition of the avg-avg objective, we have $\mathsf{cost}(c_{2k}) = \frac{1}{2}$, and $\mathsf{cost}(c_i) = \frac{5 - \frac{2}{k}}{2}$, for all $1 \leq i \leq k$. Thus, $c_{2k}$ is the optimal candidate. It follows that the distortion of mechanism $\Psi$:

| $d(\cdot,\cdot)$ | $c_i$ |
|---|---|
| $v_{2i-1}$ | 2 |
| $v_{2i}$ | 1 |
| $v_{2j-1}$ | 2 |
| $v_{2j}$ | 3 |

Table 3: For any $1 \le i, j \le k$ with $i \ne j$, the shortest-path distances in graph $G$ between candidates $c_1, c_2, \ldots, c_k$ and the voters used in the proof of Theorem 3.8.

| $d(\cdot,\cdot)$ | $c_i$ | $c_{2k}$ |
|---|---|---|
| $v_{2j-1}$ | 2 | 0 |
| $v_{2j}$ | 3 | 1 |

Table 4: For any $k+1 \le i \le 2k-1$ and $1 \le j \le k$, the shortest-path distances in graph $G$ between candidates $c_{k+1}, c_{k+2}, \ldots, c_{2k-1}, c_{2k}$ and the voters used in the proof of theorem 3.8.

$$D(\Psi) \ge \frac{\min_{1 \le i \le k}\left(\mathsf{cost}(c_i)\right)}{\mathsf{cost}(\mathsf{o})}$$

$$= \frac{\frac{5-\frac{2}{k}}{2}}{\mathsf{cost}(c_{2k})}$$

$$= 5 - \frac{2}{k}.$$

$\square$

## C  Proofs for Section 4 (Distortion Bounds for rand-rand)

**Theorem 4.1.** *For the max-max objective in general metric spaces, we have* $D((\mathsf{f}_{rd}, \mathsf{f}_{ur})) \le 3$.

*Proof.* Consider an instance $\mathcal{I} = (\mathcal{V}, \mathcal{C}, \mathcal{G}, \pi, \mathsf{d})$ and rand-rand mechanism $\Psi = (\mathsf{f}_{rd}, \mathsf{f}_{ur})$. For any voter $v \in \mathcal{V}$, we have

$$\begin{aligned}
\mathsf{cost}(\mathsf{top}(v)) &= \mathsf{d}(v^{**}(\mathsf{top}(v)), \mathsf{top}(v)) & \text{(Definition of max-max)} \\
&\le \mathsf{d}(v^{**}(\mathsf{top}(v)), \mathsf{o}) + \mathsf{d}(\mathsf{o}, \mathsf{top}(v)) & \text{(Triangle Inequality)} \\
&\le \mathsf{d}(v^{**}(\mathsf{top}(v)), \mathsf{o}) + \mathsf{d}(v, \mathsf{o}) + \mathsf{d}(v, \mathsf{top}(v)) & \text{(Triangle Inequality)} \\
&\le \mathsf{d}(v^{**}(\mathsf{o}), \mathsf{o}) + \mathsf{d}(v, \mathsf{o}) + \mathsf{d}(v, \mathsf{top}(v)) & \text{(Observation 2.9)} \\
&\le \mathsf{d}(v^{**}(\mathsf{o}), \mathsf{o}) + \mathsf{d}(v, \mathsf{o}) + \mathsf{d}(v, \mathsf{o}) & \text{(Observation 2.8)} \\
&= \mathsf{d}(v^{**}(\mathsf{o}), \mathsf{o}) + 2\mathsf{d}(v, \mathsf{o}) & \\
&\le 3\mathsf{d}(v^{**}(\mathsf{o}), \mathsf{o}) & \text{(Observation 2.9)} \\
&= 3\mathsf{cost}(\mathsf{o}) & (\mathsf{cost}(\mathsf{o}) = \mathsf{d}(v^{**}(\mathsf{o}), \mathsf{o})).
\end{aligned}$$

Combining this with Observation 2.6, we have

$$\begin{aligned}
\mathbb{E}[\mathsf{cost}(\mathsf{w})] &= \frac{1}{k}\sum_{g \in \mathcal{G}}\frac{1}{n_g}\sum_{v \in g}\mathsf{cost}(\mathsf{top}(v)) \\
&\le 3\mathsf{cost}(\mathsf{o}).
\end{aligned}$$

$\square$

**Theorem 4.2.** *For the avg-max objective in general metric spaces, we have* $D((\mathsf{f}_{rd}, \mathsf{f}_{ur})) \le 3$.

| Voter | Preference Profile |
|---|---|
| $v_{2i-1}$ | $\sigma \uparrow c_i \uparrow c_{2k}$ |
| $v_{2i}$ | $\sigma \uparrow c_{2k} \uparrow c_i$ |

Table 5: The preference profiles of the voters within each group $g_i$ $(1 \leq i \leq k)$, used in the proof of theorem 3.8.

*Proof.* Consider an instance $\mathcal{I} = (\mathcal{V}, \mathcal{C}, \mathcal{G}, \pi, \mathsf{d})$ and rand-rand mechanism $\Psi = (\mathsf{f}_{rd}, \mathsf{f}_{ur})$. By definition of the avg-max objective for any voter $v \in \mathcal{V}$, we have

$$\mathsf{cost}(\mathsf{top}(v)) = \frac{1}{k} \sum_{g \in \mathcal{G}} \mathsf{cost}_g(\mathsf{top}(v)).$$

Now, for any groups $g, g' \in \mathcal{G}$ and any voter $v \in g'$, we have

$$
\begin{aligned}
\mathsf{cost}_g(\mathsf{top}(v)) &= \mathsf{d}(v^*(\mathsf{top}(v), g), \mathsf{top}(v)) && \text{(Definition of } \mathsf{cost}_g(.)) \\
&\leq \mathsf{d}(v^*(\mathsf{top}(v), g), v) + \mathsf{d}(v, \mathsf{top}(v)) && \text{(Triangle Inequality)} \\
&\leq \mathsf{d}(v^*(\mathsf{top}(v), g), v) + \mathsf{d}(v, \mathsf{o}) && \text{(Observation 2.8)} \\
&\leq \mathsf{d}(v^*(\mathsf{top}(v), g), \mathsf{o}) + \mathsf{d}(v, \mathsf{o}) + \mathsf{d}(v, \mathsf{o}) && \text{(Triangle Inequality)} \\
&\leq \mathsf{d}(v^*(\mathsf{o}, g), \mathsf{o}) + 2\mathsf{d}(v, \mathsf{o}) && \text{(Observation 2.10)} \\
&= \mathsf{cost}_g(\mathsf{o}) + 2\mathsf{d}(v, \mathsf{o}) && \text{(Definition of } \mathsf{cost}_g(.)) \\
&\leq \mathsf{cost}_g(\mathsf{o}) + 2\mathsf{d}(v^*(\mathsf{o}, g'), \mathsf{o}) && \text{(Observation 2.10)} \\
&= \mathsf{cost}_g(\mathsf{o}) + 2\mathsf{cost}_{g'}(\mathsf{o}) && \text{(Definition of } \mathsf{cost}_g(.)).
\end{aligned}
$$

Combining this with Observation 2.6, we obtain

$$
\begin{aligned}
\mathrm{E}[\mathsf{cost}(\mathsf{w})] &= \frac{1}{k} \sum_{g \in \mathcal{G}} \frac{1}{n_g} \sum_{v \in g} \frac{1}{k} \sum_{g' \in \mathcal{G}} \mathsf{cost}_{g'}(\mathsf{top}(v)) \\
&\leq \frac{1}{k} \sum_{g \in \mathcal{G}} \frac{1}{n_g} \sum_{v \in g} \frac{1}{k} \sum_{g' \in \mathcal{G}} (2\mathsf{cost}_g(\mathsf{o}) + \mathsf{cost}_{g'}(\mathsf{o})) \\
&= \frac{1}{k} \sum_{g \in \mathcal{G}} \left( 2\mathsf{cost}_g(\mathsf{o}) + \frac{1}{k} \sum_{g' \in \mathcal{G}} \mathsf{cost}_{g'}(\mathsf{o}) \right) \\
&= \frac{1}{k} \sum_{g \in \mathcal{G}} 2\mathsf{cost}_g(\mathsf{o}) + \mathsf{cost}(\mathsf{o}) && \text{(Definition of } \mathsf{cost}(.)) \\
&= 3\mathsf{cost}(\mathsf{o}) && \text{(Definition of } \mathsf{cost}(.)).
\end{aligned}
$$

$\square$

**Theorem 4.3.** *For the max-avg objective in general metric spaces, we have* $\mathsf{D}((\mathsf{f}_{rd}, \mathsf{f}_{ur})) \leq 3$.

*Proof.* Consider an instance $\mathcal{I} = (\mathcal{V}, \mathcal{C}, \mathcal{G}, \pi, \mathsf{d})$ and rand-rand mechanism $\Psi = (\mathsf{f}_{rd}, \mathsf{f}_{ur})$. By the definition of the max-avg objective for any voter $v \in \mathcal{V}$, we have

$$
\begin{aligned}
\mathsf{cost}(\mathsf{top}(v)) &= \max_{g \in \mathcal{G}} \mathsf{cost}_g(\mathsf{top}(v)) \\
&= \mathsf{cost}_{g^*(\mathsf{top}(v))}(\mathsf{top}(v)) \\
&= \frac{1}{n_{g^*(\mathsf{top}(v))}} \sum_{v' \in g^*(\mathsf{top}(v))} \mathsf{d}(v', \mathsf{top}(v)) && \text{(Definition of } \mathsf{cost}_g(.)).
\end{aligned}
$$

Thus, we have

$$
\begin{aligned}
\mathsf{cost}_{g^*(\mathsf{top}(v))}(\mathsf{top}(v)) &= \frac{1}{n_{g^*(\mathsf{top}(v))}} \sum_{v' \in g^*(\mathsf{top}(v))} \mathsf{d}(v', \mathsf{top}(v)) \\
&\leq \frac{1}{n_{g^*(\mathsf{top}(v))}} \sum_{v' \in g^*(\mathsf{top}(v))} \left( \mathsf{d}(v, \mathsf{top}(v)) + \mathsf{d}(v, v') \right) && \text{(Triangle Inequality)}
\end{aligned}
$$

$$\leq \frac{1}{n_{g^*(\mathsf{top}(v))}} \sum_{v' \in g^*(\mathsf{top}(v))} \Big( \mathsf{d}(v,\mathsf{o}) + \mathsf{d}(v,v') \Big) \qquad \text{(Observation 2.8)}$$

$$\leq \frac{1}{n_{g^*(\mathsf{top}(v))}} \sum_{v' \in g^*(\mathsf{top}(v))} \Big( \mathsf{d}(v,\mathsf{o}) + \mathsf{d}(v,\mathsf{o}) + \mathsf{d}(\mathsf{o},v') \Big) \quad \text{(Triangle Inequality)}$$

$$= 2\mathsf{d}(v,\mathsf{o}) + \frac{1}{n_{g^*(\mathsf{top}(v))}} \sum_{v' \in g^*(\mathsf{top}(v))} \mathsf{d}(\mathsf{o},v')$$

$$= 2\mathsf{d}(v,\mathsf{o}) + \mathsf{cost}_{g^*(\mathsf{top}(v))}(\mathsf{o})$$

$$\leq 2\mathsf{d}(v,\mathsf{o}) + \mathsf{cost}(\mathsf{o}) \qquad \text{(Observation 2.7)}.$$

Combining this with Observation 2.6, we obtain

$$\mathrm{E}[\mathsf{cost}(\mathsf{w})] = \frac{1}{k} \sum_{g \in \mathcal{G}} \frac{1}{n_g} \sum_{v \in g} \mathsf{cost}(\mathsf{top}(v))$$

$$= \frac{1}{k} \sum_{g \in \mathcal{G}} \frac{1}{n_g} \sum_{v \in g} \mathsf{cost}_{g^*(\mathsf{top}(v))}(\mathsf{top}(v)) \qquad \text{(Definition of } \mathsf{cost}(.))$$

$$\leq \frac{1}{k} \sum_{g \in \mathcal{G}} \frac{1}{n_g} \sum_{v \in g} (2\mathsf{d}(v,\mathsf{o}) + \mathsf{cost}(\mathsf{o}))$$

$$= \mathsf{cost}(\mathsf{o}) + \frac{1}{k} \sum_{g \in \mathcal{G}} \frac{1}{n_g} \sum_{v \in g} 2\mathsf{d}(v,\mathsf{o})$$

$$= \mathsf{cost}(\mathsf{o}) + \frac{2}{k} \sum_{g \in \mathcal{G}} \mathsf{cost}_g(\mathsf{o}) \qquad \text{(Definition of } \mathsf{cost}_g(.))$$

$$\leq \mathsf{cost}(\mathsf{o}) + \frac{2}{k} \sum_{g \in \mathcal{G}} \mathsf{cost}(\mathsf{o}) \qquad \text{(Observation 2.7)}$$

$$= 3\mathsf{cost}(\mathsf{o}).$$

$\square$

**Theorem 4.4.** *For the avg-avg objective in general metric spaces, we have* $\mathsf{D}((\mathsf{f}_{rd}, \mathsf{f}_{ur})) \leq 3 - {}^2\!/_{kn^*}$ *where $n^*$ represents the maximum value of $n_g$ over all groups.*

*Proof.* Consider an instance $\mathcal{I} = (\mathcal{V}, \mathcal{C}, \mathcal{G}, \pi, \mathsf{d})$ and rand-rand mechanism $\Psi = (\mathsf{f}_{rd}, \mathsf{f}_{ur})$. By the definition of the avg-avg objective for any voter $v \in \mathcal{V}$, we have

$$\mathsf{cost}(\mathsf{top}(v)) = \frac{1}{k} \sum_{g' \in \mathcal{G}} \mathsf{cost}_{g'}(\mathsf{top}(v)).$$

For any voter $v \in \mathcal{V}$ and group $g'$, we have

$$\mathsf{cost}_{g'}(\mathsf{top}(v)) = \frac{1}{n_{g'}} \sum_{v' \in g'} \mathsf{d}(v', \mathsf{top}(v)) \qquad \text{(Definition of } \mathsf{cost}_g(.))$$

$$\leq \frac{1}{n_{g'}} \sum_{v' \in g'} \Big( \mathsf{d}(v, \mathsf{top}(v)) + \mathsf{d}(v', v) \Big) \qquad \text{(Triangle Inequality)}$$

$$\leq \frac{1}{n_{g'}} \sum_{v' \in g'} \Big( \mathsf{d}(v, \mathsf{o}) + \mathsf{d}(v', v) \Big) \qquad \text{(Observation 2.8)}$$

$$= \mathsf{d}(v, \mathsf{o}) + \frac{1}{n_{g'}} \sum_{v' \in g'} \mathsf{d}(v', v).$$

Combining this with Observation 2.6, we obtain

$$\mathrm{E}[\mathsf{cost}(\mathsf{w})] = \frac{1}{k} \sum_{g \in \mathcal{G}} \frac{1}{n_g} \sum_{v \in g} \frac{1}{k} \sum_{g' \in \mathcal{G}} \mathsf{cost}_{g'}(\mathsf{top}(v)) \qquad \text{(Definition of } \mathsf{cost}(.))$$

$$\leq \frac{1}{k}\sum_{g\in\mathcal{G}}\frac{1}{n_g}\sum_{v\in g}\frac{1}{k}\sum_{g'\in\mathcal{G}}\left(\mathsf{d}(v,\mathsf{o})+\frac{1}{n_{g'}}\sum_{v'\in g'}\mathsf{d}(v',v)\right)$$

$$= \frac{1}{k}\sum_{g\in\mathcal{G}}\frac{1}{n_g}\sum_{v\in g}\mathsf{d}(v,\mathsf{o})+\frac{1}{k}\sum_{g\in\mathcal{G}}\frac{1}{n_g}\sum_{v\in g}\frac{1}{k}\sum_{g'\in\mathcal{G}}\frac{1}{n_{g'}}\sum_{v'\in g'}\mathsf{d}(v',v)$$

$$= \mathsf{cost}(\mathsf{o})+\frac{1}{k}\sum_{g\in\mathcal{G}}\frac{1}{n_g}\sum_{v\in g}\frac{1}{k}\sum_{g'\in\mathcal{G}}\frac{1}{n_{g'}}\sum_{v'\in g'}\mathsf{d}(v',v) \qquad\text{(Definition of avg-avg)}$$

$$\leq \mathsf{cost}(\mathsf{o})+\frac{1}{k}\sum_{g\in\mathcal{G}}\frac{1}{n_g}\sum_{v\in g}\frac{1}{k}\sum_{g'\in\mathcal{G}}\frac{1}{n_{g'}}\sum_{v'\in g',\,v'\neq v}\left(\mathsf{d}(v,\mathsf{o})+\mathsf{d}(\mathsf{o},v')\right) \qquad\text{(Triangle Inequality)}$$

$$= \mathsf{cost}(\mathsf{o})+\frac{1}{k}\sum_{g\in\mathcal{G}}\frac{1}{n_g}\sum_{v\in g}\frac{1}{k}\sum_{g'\in\mathcal{G}}\frac{1}{n_{g'}}\sum_{v'\in g'}\left(\mathsf{d}(v,\mathsf{o})+\mathsf{d}(\mathsf{o},v')\right)$$

$$- \frac{1}{k}\sum_{g\in\mathcal{G}}\frac{1}{n_g}\sum_{v\in g}\frac{1}{k}\sum_{g'\in\mathcal{G},\,g'=g}\frac{1}{n_{g'}}\sum_{v'\in g',\,v'=v}\left(\mathsf{d}(v,\mathsf{o})+\mathsf{d}(\mathsf{o},v')\right)$$

$$= \mathsf{cost}(\mathsf{o})+\frac{1}{k}\sum_{g\in\mathcal{G}}\frac{1}{n_g}\sum_{v\in g}\frac{1}{k}\sum_{g'\in\mathcal{G}}\frac{1}{n_{g'}}\sum_{v'\in g'}\left(\mathsf{d}(v,\mathsf{o})+\mathsf{d}(\mathsf{o},v')\right)$$

$$- \frac{1}{k}\sum_{g\in\mathcal{G}}\frac{1}{n_g}\sum_{v\in g}\frac{1}{k}\cdot\frac{1}{n_g}\cdot 2\mathsf{d}(v,\mathsf{o})$$

$$= \mathsf{cost}(\mathsf{o})+2\mathsf{cost}(\mathsf{o})-\frac{1}{k}\sum_{g\in\mathcal{G}}\frac{1}{n_g}\sum_{v\in g}\frac{1}{k}\cdot\frac{1}{n_g}\cdot 2\mathsf{d}(v,\mathsf{o}) \qquad\text{(Definition of avg-avg)}$$

$$= 3\mathsf{cost}(\mathsf{o})-\frac{1}{k}\sum_{g\in\mathcal{G}}\frac{1}{n_g}\sum_{v\in g}\frac{1}{k}\cdot\frac{1}{n_g}\cdot 2\mathsf{d}(v,\mathsf{o})$$

$$= 3\mathsf{cost}(\mathsf{o})-\frac{2}{k^2}\sum_{g\in\mathcal{G}}\frac{1}{n_g}\cdot\mathsf{cost}_g(\mathsf{o}) \qquad\text{(Definition of } \mathsf{cost}_g(.))$$

$$\leq 3\mathsf{cost}(\mathsf{o})-\frac{2}{k^2}\sum_{g\in\mathcal{G}}\frac{1}{n^*}\cdot\mathsf{cost}_g(\mathsf{o})$$

$$= (3-\frac{2}{kn^*})\mathsf{cost}(\mathsf{o}) \qquad\text{(Definition of avg-avg)}.$$

$\square$

**Theorem 4.6.** *For the* max-avg *and* max-max *objectives, the distortion of any* rand-rand *mechanism is at least* 3, *even when the metric space is a line.*

*Proof.* Consider any rand-rand mechanism $\Psi = (\mathsf{f}_{in}, \mathsf{f}_{ov})$. We construct an instance with candidates $\mathcal{C} = \{c_1, c_2, c_3\}$ and voters $\mathcal{V} = \{v_1, v_2\}$, where each voter belongs to a distinct group: $v_1 \in g_1$ and $v_2 \in g_2$. The instance is constructed on the line metric as follows (refer to Figure 6):

- Candidates $c_1$, $c_2$, and $c_3$ are located at positions $-1$, $0$, and $1$, respectively.

- Voters $v_1$ and $v_2$ are located at positions $-0.5$ and $0.5$, respectively.

- The preference profile of each voter is $\pi_1 = (c_1, c_2, c_3)$ for $v_1$ and $\pi_2 = (c_3, c_2, c_1)$ for $v_2$.

Trivially, the preference profiles are consistent with the distances in Figure 6. According to Observation 2.1, candidates $c_1$ and $c_3$ are chosen as the representatives of groups $g_1$ and $g_2$, respectively, and then mechanism $\Psi$ must select one of them as the final winner. We have $\mathsf{cost}(c_1) = \mathsf{cost}(c_3) = \frac{3}{2}$ and $\mathsf{cost}(c_2) = \frac{1}{2}$. Clearly, $c_2$ is the optimal candidate. The distortion of $\Psi$ is:

$$\mathsf{D}(\Psi) \geq \frac{\min\big(\mathsf{cost}(c_1), \mathsf{cost}(c_3)\big)}{\mathsf{cost}(\mathsf{o})}$$

Figure 6: An example used in the proof of Theorem 4.6.

Figure 7: Configuration of the candidates and voters in instance $\mathcal{I}_i$ used in the proof of Theorem 4.7.

$$= \frac{\frac{3}{2}}{\mathsf{cost}(c_2)}$$
$$= 3.$$

$\square$

**Theorem 4.7.** *For the* avg-max *objective, the distortion of any* rand-rand *mechanism is at least* $3 - \frac{2}{n}$ *(equivalently,* $3 - \frac{2}{m}$ *), even when the metric space is a line.*

*Proof.* We construct an instance with a set of $m$ candidates, $\mathcal{C} = \{c_1, c_2, \ldots, c_m\}$, and a set of $n = m$ voters, $\mathcal{V} = \{v_1, v_2, \ldots, v_n\}$, all belonging to a single group. Each voter $v_i$ has a preference profile $\pi_i$ that ranks candidates cyclically, starting with $c_i$ as their top choice:

$$\pi_i = (c_i, c_{i+1}, \ldots, c_m, c_1, c_2, \ldots, c_{i-1}).$$

Now, we construct $m$ instances, denoted $\mathcal{I}_1, \mathcal{I}_2, \ldots, \mathcal{I}_m$, on the line metric. Across all instances, the voter set $\mathcal{V}$, candidate set $\mathcal{C}$, and the preference profile $\pi$ are identical; they differ only in the arrangement of voters and candidates within the underlying metric space.

For any instance $\mathcal{I}_i$, where $1 \le i \le m$, configuration of the line metric is as follows (see Figure 7):

- Candidate $c_i$ is located at position $0$, while all other candidates are located at $1$.

- Voter $v_i$ is located at position $-0.5$, and all other voters are located at $0.5$.

It is straightforward to verify that each constructed instance is consistent with the specified preference profile. Since there is a single group, the avg-max objective simplifies to max. For each instance $\mathcal{I}_i$, we have $\mathsf{cost}(c_i) = \frac{1}{2}$, and $\mathsf{cost}(c_j) = \frac{3}{2}$, for $1 \le j \le m$ (where $i \ne j$). Clearly, $c_i$ is the optimal candidate in $\mathcal{I}_i$.

Now, consider any rand-rand mechanism $\Psi$. Let $p_i$ denote the probability that $\Psi$ selects $c_i$ as the winner, where $\sum_{i=1}^{m} p_i = 1$. For instance $\mathcal{I}_i$, the mechanism's expected cost is

$$\mathrm{E}[\mathsf{cost}(\Psi(\mathcal{I}_i))] = \sum_{j=1}^{m} p_j \cdot \mathsf{cost}(c_j) = \sum_{j \ne i} p_j \cdot \frac{3}{2} + p_i \cdot \frac{1}{2} = \frac{3}{2}(1 - p_i) + \frac{p_i}{2} = \frac{3}{2} - p_i.$$

It follows that

$$\frac{\mathrm{E}[\mathsf{cost}(\Psi(\mathcal{I}_i))]}{\mathsf{cost}(\mathsf{o})} = \frac{\frac{3}{2} - p_i}{\frac{1}{2}} = 3 - 2p_i.$$

Since the total probability must sum up to 1, there exists some index $i$ such that $p_i \le \frac{1}{m} = \frac{1}{n}$. Finally, we obtain the following lower bound on the distortion of $\Psi$

$$\mathsf{D}(\Psi) \ge 3 - \frac{2}{m} = 3 - \frac{2}{n}.$$

$\square$

**Theorem 4.9.** *For general metric spaces and the* avg-avg *objective, the distortion of any* rand-rand *mechanism is at least* $3 - \frac{2}{n}$ *(equivalently* $3 - \frac{2}{k}$ *).*

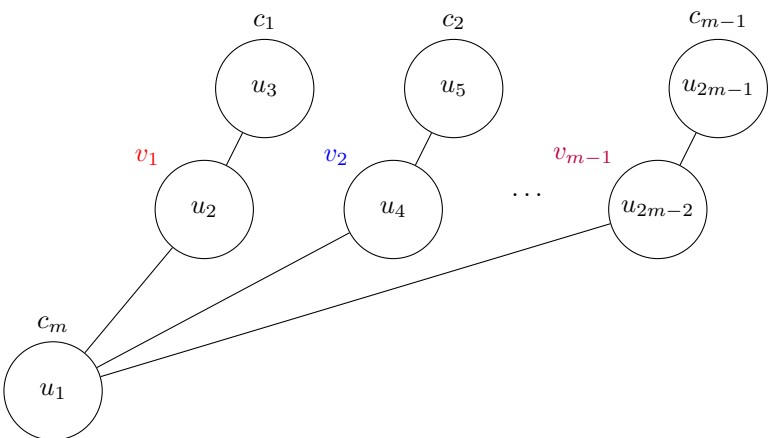

Figure 8: Tree graph used in the proof of Theorem 4.9. Different voter groups are distinguished by distinct colors.

| $d(\cdot, \cdot)$ | $c_i$ | $c_j$ | $c_m$ |
|---|---|---|---|
| $v_i$ | 1 | 3 | 1 |

Table 6: For any $1 \le i, j \le k$ with $i \ne j$, the shortest-path distances between candidates and voters, derived from tree graph $G$ in the proof of Theorem 4.9.

*Proof.* Consider a rand-rand mechanism $\Psi$. We construct an instance with a set of $m = k + 1$ candidates, $\mathcal{C} = \{c_1, c_2, \ldots, c_{m=k+1}\}$, a set of $n = k$ voters $\mathcal{V} = \{v_1, v_2, \ldots, v_{n=k}\}$, and $k$ single-voter groups, $g_i = \{v_i\}$ for $1 \le i \le k$. Each voter $v_i$ has a preference profile $\pi_i$, with $c_i$ as the top choice, immediately followed by $c_m$, denoted as $\pi_i = \sigma \uparrow c_m \uparrow c_i$, where $\sigma$ is an ordering of the candidates.

We now construct a connected graph $G$ with $n + m$ vertices, denoted $u_1, u_2, \ldots, u_{n+m}$, where the shortest-path distances in $G$ define the underlying metric space $d$. Each voter and candidate is placed on one of the vertices. The construction of $G$ is as follows (see Figure 8 for an illustration):

- Place candidate $c_m$ at vertex $u_1$.

- For each $1 \le i \le k$, place voter $v_i$ at vertex $u_{2i}$, and candidate $c_i$ at vertex $u_{2i+1}$.

- For each $1 \le i \le k$, add an edge between $u_1$ and $u_{2i}$, and another edge between $u_{2i}$ and $u_{2i+1}$. This forms $k$ branches extending from the central vertex $u_1$.

See Table 6 for the corresponding distances. For all $1 \le i \le k$, we have $\text{top}(v_i) = c_i$. Therefore, the representative of group $g_i$ is candidate $c_i$ (by Observation 2.1). Consequently $c_m$ is not the representative of any group, and thus cannot be the winner of mechanism $\Psi$. According to the definition of the avg-avg objective, we have $\text{cost}(c_i) = \frac{3(n-1)+1}{n} = 3 - \frac{2}{n}$ for $1 \le i < m$, and $\text{cost}(c_m) = 1$. Clearly, $c_m$ is the optimal candidate. We obtain the lower bound on the distortion of $\Psi$ as follows:

$$
\begin{aligned}
D(\Psi) &\ge \min_{1 \le i < m} \left( \frac{\text{cost}(c_i)}{\text{cost}(o)} \right) \\
&= \frac{3 - \frac{2}{n}}{\text{cost}(c_m)} \\
&= 3 - \frac{2}{n}.
\end{aligned}
$$

$\square$

# D    Proofs for Section 5 (Resolving the Distortion Bounds for **det-det**)

**Theorem 5.1.** *For the max-max objective in general metric spaces, we have* $\mathsf{D}(\mathsf{m}_{ad}) \leq 3$.

*Proof.* Consider an instance $\mathcal{I} = (\mathcal{V}, \mathcal{C}, \mathcal{G}, \pi, \mathsf{d})$ and the *Arbitrary Dictator* mechanism ($\Psi = \mathsf{m}_{ad}$), which selects the top-ranked candidate of an arbitrary voter $v$ as the final winner; that is, $\mathsf{w} = \mathsf{top}(v)$. We follow a strategy roughly analogous to that in Theorem 4.1.

$$
\begin{aligned}
\mathsf{cost}(\mathsf{top}(v)) &= \mathsf{d}(v^{**}(\mathsf{top}(v)), \mathsf{top}(v)) && \text{(Definition of max-max)} \\
&\leq \mathsf{d}(v^{**}(\mathsf{top}(v)), v) + \mathsf{d}(v, \mathsf{top}(v)) && \text{(Triangle Inequality)} \\
&\leq \mathsf{d}(v^{**}(\mathsf{top}(v)), v) + \mathsf{d}(v, \mathsf{o}) && \text{(Observation 2.8)} \\
&\leq \mathsf{d}(v^{**}(\mathsf{top}(v)), \mathsf{o}) + \mathsf{d}(\mathsf{o}, v) + \mathsf{d}(v, \mathsf{o}) && \text{(Triangle Inequality)} \\
&= \mathsf{d}(v^{**}(\mathsf{top}(v)), \mathsf{o}) + 2\mathsf{d}(v, \mathsf{o}) && \\
&\leq \mathsf{d}(v^{**}(\mathsf{top}(v)), \mathsf{o}) + 2\mathsf{d}(v^{**}(\mathsf{o}), \mathsf{o}) && \text{(Observation 2.9)} \\
&\leq 3\mathsf{d}(v^{**}(\mathsf{o}), \mathsf{o}) && \text{(Observation 2.9)} \\
&= 3\mathsf{cost}(\mathsf{o}) && \text{(Definition of max-max).}
\end{aligned}
$$

$\square$

**Theorem 5.2.** *For the avg-max objective in general metric spaces, we have* $\mathsf{D}((\mathsf{f}_{par}, \mathsf{f}_\beta)) \leq 2\beta + 3$.

*Proof.* Consider an instance $\mathcal{I} = (\mathcal{V}, \mathcal{C}, \mathcal{G}, \pi, \mathsf{d})$ and a det-det mechanism $\Psi = (\mathsf{f}_{par}, \mathsf{f}_\beta)$. By the property of Pareto efficiency, for each group $g$, there exists a voter $v_g \in g$ who prefers the representative $\mathsf{w}_g$ over the optimal candidate $\mathsf{o}$. We have

$$
\begin{aligned}
\mathsf{cost}(\mathsf{w}) &= \frac{1}{k} \sum_{g \in \mathcal{G}} \mathsf{d}(v^*(\mathsf{w}, g), \mathsf{w}) && \text{(Definition of avg-max)} \\
&\leq \frac{1}{k} \sum_{g \in \mathcal{G}} \mathsf{d}(v^*(\mathsf{w}, g), \mathsf{o}) + \mathsf{d}(\mathsf{o}, \mathsf{w}) && \text{(Triangle Inequality)} \\
&\leq \frac{1}{k} \sum_{g \in \mathcal{G}} \mathsf{d}(v^*(\mathsf{o}, g), \mathsf{o}) + \mathsf{d}(\mathsf{o}, \mathsf{w}) && \text{(Observation 2.10)} \\
&= \mathsf{cost}(\mathsf{o}) + \frac{1}{k} \sum_{g \in \mathcal{G}} \mathsf{d}(\mathsf{o}, \mathsf{w}) && \text{(Definition of avg-max)} \\
&\leq \mathsf{cost}(\mathsf{o}) + \frac{1}{k} \sum_{g \in \mathcal{G}} \mathsf{d}(\mathsf{o}, \mathsf{w}_g) + \mathsf{d}(\mathsf{w}, \mathsf{w}_g) && \text{(Triangle Inequality)} \\
&\leq \mathsf{cost}(\mathsf{o}) + (\beta + 1) \cdot \frac{1}{k} \sum_{g \in \mathcal{G}} \mathsf{d}(\mathsf{o}, \mathsf{w}_g) && \text{(Observation 2.12)} \\
&\leq \mathsf{cost}(\mathsf{o}) + (\beta + 1) \cdot \frac{1}{k} \sum_{g \in \mathcal{G}} \mathsf{d}(\mathsf{o}, v_g) + \mathsf{d}(v_g, \mathsf{w}_g) && \text{(Triangle Inequality)} \\
&\leq \mathsf{cost}(\mathsf{o}) + 2(\beta + 1) \cdot \frac{1}{k} \sum_{g \in \mathcal{G}} \mathsf{d}(\mathsf{o}, v_g) && (\mathsf{d}(v_g, \mathsf{w}_g) \leq \mathsf{d}(v_g, \mathsf{o})) \\
&\leq \mathsf{cost}(\mathsf{o}) + 2(\beta + 1) \cdot \frac{1}{k} \sum_{g \in \mathcal{G}} \mathsf{d}(\mathsf{o}, v^*(\mathsf{o}, g)) && \text{(Observation 2.10)} \\
&= (2\beta + 3)\mathsf{cost}(\mathsf{o}) && \text{(Definition of avg-max).}
\end{aligned}
$$

$\square$

**Theorem 5.4.** *For general metric spaces and the max-avg objective, the distortion of any det-det mechanism is at least* $5$.

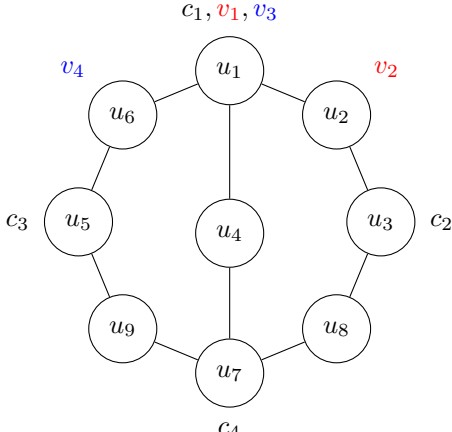

Figure 9: An illustration of the graph $G$ used in the proof of Theorem 5.4. Different voter groups are distinguished by distinct colors.

| $d(\cdot, \cdot)$ | $c_1$ | $c_2$ | $c_3$ | $c_4$ |
|---|---|---|---|---|
| $v_1$ | 0 | 2 | 2 | 2 |
| $v_2$ | 1 | 1 | 3 | 3 |
| $v_3$ | 0 | 2 | 2 | 2 |
| $v_4$ | 1 | 3 | 1 | 3 |

Table 7: The shortest-path distances between candidates and voters, as derived from the graph $G$ in the proof of theorem 5.4.

*Proof.* Consider a det-det mechanism $\Psi = (f_{in}, f_{ov})$. We construct an instance with a set of candidates $\mathcal{C} = \{c_1, c_2, c_3, c_4\}$, a set of voters $\mathcal{V} = \{v_1, v_2, v_3, v_4\}$, and 2 groups $g_1 = \{v_1, v_2\}$ and $g_2 = \{v_3, v_4\}$. Let $\sigma$ be an arbitrary ordering of the candidates. Without loss of generality, assume that $c_1$ has an in-degree of at least $\left\lceil \frac{m-1}{2} \right\rceil = 2$ in the tournament $\mathcal{T}(f_{in}, \mathcal{C}, \sigma)$, such a candidate is guaranteed to exist by Observation 2.2. Further, let $c_2$ and $c_3$ be the two candidates with directed edges toward $c_1$ in the tournament.

We now construct a connected graph $G$ with 9 vertices, denoted $u_1, u_2, \ldots u_9$, where the shortest-path distances in $G$ define the underlying metric space $d$. Each voter and candidate is placed at one of the vertices (a single vertex may host multiple entities). The graph configuration is as shown in Figure 9.

Pairwise distances between the candidates and voters are presented in Table 7. Now, we define the preference profiles in Table 8, which are consistent with both the shortest-path distances in graph $G$ and the input of tournament $\mathcal{T}(f_{in}, \mathcal{C}, \sigma)$. Note that multiple profiles may be consistent.

Since candidates $c_2$ and $c_3$ defeat $c_1$ in $\mathcal{T}(f_{in}, \mathcal{C}, \sigma)$, they must serve as the representative of groups $g_1$ and $g_2$, respectively. Thus, the set of representatives is $R = \{c_2, c_3\}$.

At the second stage of the distributed voting process, we consider two instances:

- $\mathcal{I}_1$: The preference profiles of $c_2$ and $c_3$ are $(c_2, c_1, c_4, c_3)$ and $(c_3, c_1, c_4, c_2)$, respectively.

- $\mathcal{I}_2$: The preference profiles of $c_2$ and $c_3$ are $(c_2, c_4, c_1, c_3)$ and $(c_3, c_4, c_1, c_2)$, respectively.

It is straightforward to verify that both $\mathcal{I}_1$ and $\mathcal{I}_2$ are consistent with the metric space defined earlier. In both cases, the preference profiles of $c_2$ and $c_3$ follow the pattern

$$(c_2, c_a, c_b, c_3) \quad \text{and} \quad (c_3, c_a, c_b, c_2),$$

where $(c_a, c_b)$ is some ordering of $(c_1, c_4)$.

If the rule selects the first-ranked or fourth-ranked candidate, then $c_1$ is not chosen. If the rule selects the second-ranked candidate ($c_a$), then in $\mathcal{I}_2$ we again exclude $c_1$. If the rule selects the third-ranked

| Voter | Preference Profile |
|-------|-------------------|
| $v_1$ | $\sigma \uparrow c_2 \uparrow c_1$ |
| $v_2$ | $\sigma \uparrow c_1 \uparrow c_2$ |
| $v_3$ | $\sigma \uparrow c_3 \uparrow c_1$ |
| $v_4$ | $\sigma \uparrow c_1 \uparrow c_3$ |

Table 8: The voter preference profiles used in the proof of theorem 5.4.

candidate ($c_b$), then in $\mathcal{I}_1$ we exclude $c_1$. Therefore, in every case, there exists an instance in which the mechanism selects the winner from the set $\{c_2, c_3, c_4\}$.

By the definition of the max-avg objective, we have $\mathsf{cost}(c_1) = \frac{1}{2}$ and $\mathsf{cost}(c_2) = \mathsf{cost}(c_3) = \mathsf{cost}(c_4) = \frac{5}{2}$. Thus, $c_1$ is the optimal candidate. The distortion of mechanism $\Psi$ is obtained as follows:

$$\mathsf{D}(\Psi) \geq \min\left(\frac{\mathsf{cost}(c_2), \mathsf{cost}(c_3), \mathsf{cost}(c_4)}{\mathsf{cost}(\mathsf{o})}\right)$$

$$\geq \frac{\frac{5}{2}}{\mathsf{cost}(c_1)}$$

$$= 5.$$

□

# E  Proofs for Section 6 (An Extension of Lower Bounds for rand-rand and rand-det)

**Theorem 6.1.** *For the avg-avg objective in Euclidean space, the distortion of any rand-rand mechanism is at least $\sqrt{5} - \varepsilon$, for every constant $\varepsilon > 0$.*

*Proof.* Consider any rand-rand mechanism $\Psi$. Let $l$ be a positive integer. Consider an instance in $(l+1)-$dimensional Euclidean space, $\mathbb{R}^{l+1}$, with $l+2$ candidates, denoted $c_1, c_2, \ldots, c_{l+2}$, and $k = l+1$ groups, each with a single voter $v_i$ for $1 \leq i \leq l+1$. We construct the instance as follows:

- Let $q_i$ be the point in $\mathbb{R}^{l+1}$ whose $i$-th coordinate is 1 and all other coordinates are 0, for $1 \leq i \leq l+1$.

- Place candidate $c_i$ at point $q_i$ for each $1 \leq i \leq l+1$.

- The final candidate, $c_{l+2}$, is placed at the centroid of the other candidates; $\left(\frac{1}{l+1}, \frac{1}{l+1}, \ldots, \frac{1}{l+1}\right)$.

- In the $i$-th group, the single voter $v_i$ is positioned at the midpoint between their corresponding candidate, $c_i$ and the centroid candidate, $c_{l+2}$. Indeed, each voter $v_i$ is located at a point where the $i$-th coordinate is $\frac{l+2}{2(l+1)}$ and all other coordinates are $\frac{1}{2(l+1)}$. Note that each voter's preference profile is structured so that the top-ranked candidate of voter $v_i$ is $c_i$, consistent with the underlying Euclidean space.

In particular, in 3D space ($l = 2$), the instance lies within an equilateral triangle with vertices $(1, 0, 0)$, $(0, 1, 0)$, and $(0, 0, 1)$, as illustrated in Figure 10.

For all $1 \leq i \leq l+1$, we have

$$\mathsf{d}(c_i, v_i) = \mathsf{d}(c_{l+2}, v_i)$$

$$= \sqrt{\left(\frac{1}{2(l+1)}\right)^2 l + \left(\frac{l}{2(l+1)}\right)^2}$$

$$= \sqrt{\frac{l}{4(l+1)}}.$$

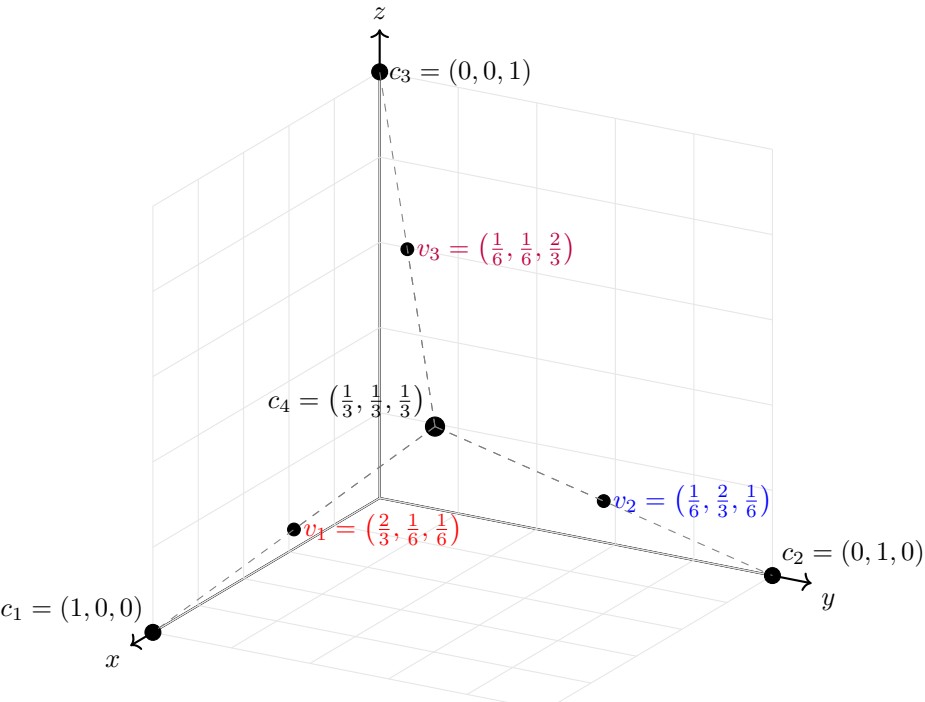

Figure 10: A $3-dimensional$ Euclidean model ($l = 2$) illustrating the geometric lower bound construction for the rand-rand mechanisms under the avg-avg objective. Candidates $c_1$, $c_2$, and $c_3$ are located at the unit basis vectors, with $c_4 = \left(\frac{1}{3}, \frac{1}{3}, \frac{1}{3}\right)$ at the centroid of the triangle they form. Voters $v_1$, $v_2$, and $v_3$ are positioned at the midpoints between the centroid $c_4$ and their top-ranked candidate $c_i$. Different group voters are indicated via distinct colors.

Moreover, for all $1 \leq i, j \leq l + 1$ (where $i \neq j$), we have

$$\mathsf{d}(c_i, v_j) = \sqrt{\left(\frac{1}{2(l+1)}\right)^2 (l-1) + \left(\frac{l+2}{2(l+1)}\right)^2 + \left(\frac{2l+1}{2l+2}\right)^2}$$

$$= \sqrt{\frac{5l+4}{4l+4}}.$$

By the definition of the avg-avg objective, the cost of each candidate is the average distance to all $l + 1$ voters. Thus, we conclude that

$$\mathsf{cost}(c_{l+2}) = \frac{1}{2}\sqrt{\frac{l}{l+1}},$$

$$\mathsf{cost}(c_i) = \frac{l\sqrt{\frac{5l+4}{4l+4}} + \frac{1}{2}\sqrt{\frac{l}{l+1}}}{l+1} \quad (1 \leq i \leq l+1).$$

Clearly, the optimal candidate is $c_{l+2}$. According to Observation 2.1, the representative of the $i$-th group is candidate $c_i$. Finally, the mechanism selects the winner from among the candidates $c_1, c_2, \ldots, c_{l+1}$. A lower bound on the distortion of the mechanism $\Psi$ is obtained as follows ($1 \leq i \leq l+1$):

$$\mathsf{D}(\Psi) \geq \frac{\mathsf{cost}(c_i)}{\mathsf{cost}(c_{l+2})}$$

$$= \frac{\frac{l}{l+1}\sqrt{\frac{5l+4}{4l+4}} + \frac{1}{2(l+1)}\sqrt{\frac{l}{l+1}}}{\frac{1}{2}\sqrt{\frac{l}{l+1}}}.$$

As $l \to \infty$, the ratio approaches $\sqrt{5} \approx 2.236$. Therefore, for any $\varepsilon > 0$, we can construct an instance with distortion greater than $\sqrt{5} - \varepsilon$. □

**Theorem 6.2.** *For the* avg-avg *objective in Euclidean space, the distortion of any* rand-det *mechanism is at least* $2 + \sqrt{5} - \varepsilon$, *for every constant* $\varepsilon > 0$.

*Proof.* Consider a rand-det mechanism $\Psi = (f_{in}, f_{ov})$, a set of $2m$ candidates $\mathcal{C} = \{c_1, c_2, \ldots, c_{2m}\}$, and an arbitrary ordering $\sigma$ over them. By Observation 2.2, the tournament $\mathcal{T}(f_{in}, \mathcal{C}, \sigma)$, must have a candidate with in-degree at least $\lceil \frac{2m-1}{2} \rceil = m$. Without loss of generality, let $c_{m+1}$ be such a candidate and let $c_1, c_2, \ldots, c_m$ be $m$ candidates that have directed edges toward $c_{m+1}$ in the tournament. We now construct the following instance with $k = m$ groups in $(m+1)$−dimensional Euclidean space:

- Let $q_i$ be the point in $\mathbb{R}^{m+1}$ whose $i$-th coordinate is 1 and all other coordinates are 0, for $1 \leq i \leq m+1$.

- Place candidate $c_i$ at point $q_i$ for each $1 \leq i \leq m$ and candidate $c_{m+1}$ at the centroid $\left( \frac{1}{m+1}, \frac{1}{m+1}, \ldots, \frac{1}{m+1} \right)$.

- Place candidate $c_i$ at point $q_{m+1}$ for each $m+2 \leq i \leq 2m$.

- In the $i$-th group ($1 \leq i \leq m$), there are two voters:
  1. Voter $v_{2i-1}$ is located at the centroid, which is the same position as candidate $c_{m+1}$.
  2. Voter $v_{2i}$ is located exactly at the midpoint between candidates $c_{m+1}$ and $c_i$, with coordinates equal to $\frac{m+2}{2(m+1)}$ in the $i$-th dimension and $\frac{1}{2(m+1)}$ in all other dimensions.

- The ordinal preferences of the $v_{2i-1}$ and $v_{2i}$ are defined as $\pi_{2i-1} = \sigma \uparrow c_i \uparrow c_{m+1}$ and $\pi_{2i} = \sigma \uparrow c_{m+1} \uparrow c_i$. These preferences are consistent with the underlying Euclidean space:
  1. The distance from $v_{2i-1}$ to all candidates except $c_{m+2}$ is equal.
  2. The distance from $v_{2i-1}$ to $c_{m+2}$ is zero.
  3. Voter $v_{2i}$ is closer to candidates $c_{m+1}$ and $c_i$ than to any other candidates, and is equidistant from all remaining ones.
  4. The distance from $v_{2i}$ to candidates $c_{m+1}$ and $c_i$ is equal.

When $m = 2$ the instance lies within an equilateral triangle with vertices at $(1, 0, 0)$, $(0, 1, 0)$, and $(0, 0, 1)$, as shown in Figure 11. For all $1 \leq i, j \leq m$, we have

$$d(c_i, c_{m+1}) = d(c_i, v_{2j-1})$$
$$= \sqrt{\left( \frac{1}{m+1} \right)^2 m + \left( \frac{m}{m+1} \right)^2}$$
$$= \sqrt{\frac{m}{m+1}},$$

for all $1 \leq i, j \leq m$ (where $i \neq j$), we have

$$d(c_i, v_{2j}) = \sqrt{\left( \frac{1}{2(m+1)} \right)^2 (m-1) + \left( \frac{m+2}{2(m+1)} \right)^2 + \left( \frac{2m+1}{2m+2} \right)^2}$$
$$= \sqrt{\frac{5m+4}{4m+4}},$$

and for all $1 \leq i \leq m$, we have

$$d(c_i, v_{2i}) = \sqrt{\left( \frac{1}{2(m+1)} \right)^2 m + \left( \frac{m}{2(m+1)} \right)^2}$$
$$= \sqrt{\frac{m}{4(m+1)}}.$$

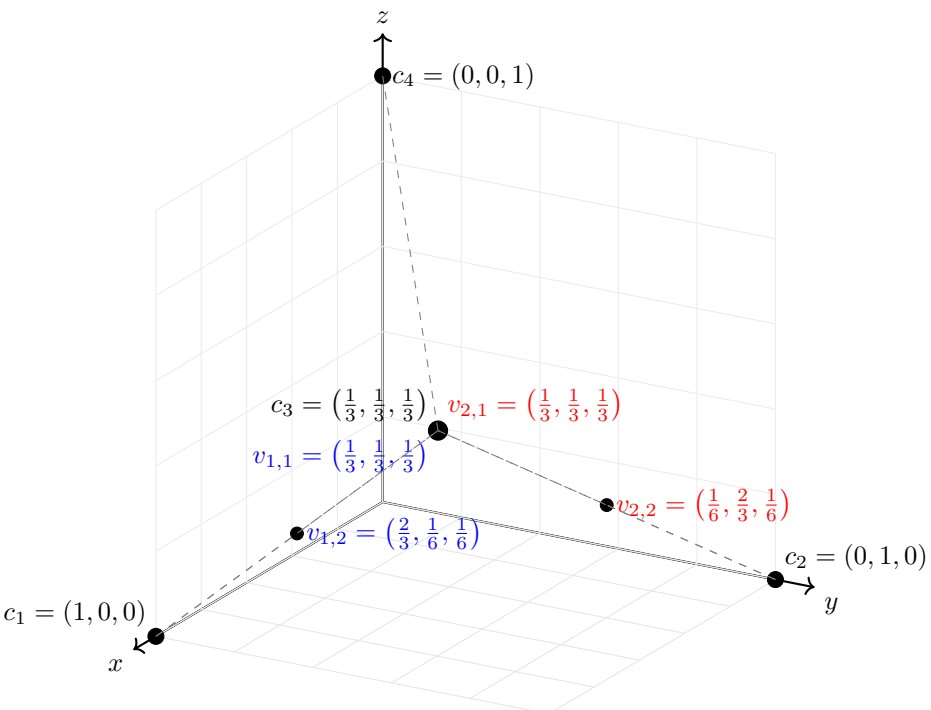

Figure 11: An illustration of the constructed instance when $m = 2$. The candidates are positioned at the corners and centroid of the 3D simplex (i.e., the equilateral triangle embedded in $\mathbb{R}^3$). Candidate $c_3$ is placed at the centroid, representing the candidate with high in-degree in $\mathcal{T}(f_{in}, \mathcal{C}, \sigma)$. Each group contains two voters: $v_{2i-1}$ is located at the centroid, while $v_{2i}$ is placed at the midpoint between $c_3$ and $c_i$ for $i = 1, 2$. Different group voters are indicated via distinct colors.

By the definition of the avg-avg objective, we conclude that

$$\mathsf{cost}(c_{m+1}) = \frac{m\left(\frac{0+\sqrt{\frac{m}{4(m+1)}}}{2}\right)}{m} = \frac{1}{4}\sqrt{\frac{m}{m+1}},$$

$$\mathsf{cost}(c_i) = \frac{\frac{\sqrt{\frac{m}{m+1}}+\sqrt{\frac{5m+4}{4m+4}}}{2}(m-1) + \frac{\sqrt{\frac{m}{m+1}}+\sqrt{\frac{m}{4(m+1)}}}{2}}{m} \qquad (1 \le i \le m).$$

Clearly, the optimal candidate is $c_{m+1}$. By the definition of $\mathcal{T}(f_{in}, \mathcal{C}, \sigma)$, the representative of group $i$ ($1 \le i \le m$) is $c_i$. Therefore, the mechanism selects the final winner from among the candidates $c_1, c_2, \ldots, c_m$. A lower bound on the distortion of the mechanism $\Psi$ is obtained as follows ($1 \le i \le m$):

$$\mathsf{D}\left(\Psi\right) \ge \frac{\mathsf{cost}(c_i)}{\mathsf{cost}(c_{m+1})}$$

$$= 2 + \frac{m-1}{m}\sqrt{\frac{5m+4}{m}} + \frac{1}{m} \qquad (1 \le i \le m),$$

As $m \to \infty$, the ratio approaches $2 + \sqrt{5} \approx 4.236$. Therefore, for any $\varepsilon > 0$, we can construct an instance with distortion greater than $2 + \sqrt{5} - \varepsilon$. $\qquad\square$

