# OpenReview forum: "Tight Bounds on the Distortion of Randomized and Deterministic Distributed Voting"
_NeurIPS.cc/2025/Conference — NeurIPS 2025 poster_

### Official Review · Reviewer_buyo · 2025-06-20

**Clarity:** 3
**Significance:** 3
**Originality:** 3
**Rating:** 4
**Confidence:** 4

**Summary:**

This paper presents a comprehensive study of metric distortion in distributed voting systems, where voters are grouped, and local winners are selected before a final aggregation. The authors consider both deterministic and randomized voting rules and analyze their performance across four natural cost objectives: avg-avg, avg-max, max-avg, and max-max. Significant theoretical advances are made, including tight or nearly tight bounds for all objectives in both rand-det and rand-rand settings. For example, the paper improves the deterministic upper bound for avg-max from 11 to 7, establishes a tight lower bound of 5 for max-avg, and proves distortion bounds of 3 for several randomized objectives. The analysis introduces novel tools, such as the Bias Tournament, and explores both general metric and Euclidean spaces. The results offer a near-complete characterization of distortion in this setting and highlight the benefits and limitations of randomization in distributed social choice. The paper is well-structured and clearly written, with detailed proofs and illustrative examples that support its contributions, making it a valuable reference for future research in metric voting theory and distributed decision-making.

**Questions:**

Why didn't you consider the det-rand mechanism?

**Ethical Concerns:**

["NO or VERY MINOR ethics concerns only"]

**Final Justification:**

The paper makes substantial progress in understanding the distortion of distributed voting mechanisms.

**Limitations:**

Yes

**Paper Formatting Concerns:**

No.

**Quality:**

3

**Strengths And Weaknesses:**

Strengths
The paper makes substantial progress in understanding the distortion of distributed voting mechanisms, closing gaps in existing bounds and introducing novel techniques like the Bias Tournament for lower bounds.

Weaknesses:
1.Authors may refine their descriptions, such as some proof sketches, in order to include a conclusion section.
2.Authors do not propose new mechanisms.

---

> ### Author Rebuttal · Authors · 2025-07-31
>
> Thank you for taking the time to review our paper. We noticed that the 'Strengths' section of the review did not align with the content of our submission and might have been included by mistake. Below, we address the technical comments.
>
> >**However, the paper’s technical exposition could be improved. While the core algorithmic ideas and analysis are sound, several important definitions and procedural steps are presented in a terse or abstract way, making the work less accessible to readers unfamiliar with the topic. For example, the intuition behind the penalty functions and dual constructions could be elaborated with more concrete examples or illustrations. Strengthening the clarity and pedagogical structure of the technical sections—particularly the algorithm descriptions and analytical proofs—would make the paper more readable and broaden its impact beyond the expert audience.**
>
> Due to space limitations, some of the definitions and algorithms were presented more tersely than we would have preferred. In the final version, we will do our best to improve the exposition, and ensure the presentation is clearer and easier to follow. We would greatly appreciate any specific feedback on definitions that may require further clarification, so we can enhance them accordingly.
>
> >**Why didn't you consider the det-rand mechanism?**
>
> Analyzing the det-rand model is indeed a promising direction for future work. The randomized nature of in-group selection limits the direct applicability of techniques such as the Bias Tournament. Due to time and space constraints, we focused on the det-det, rand-rand, and rand-det models in this work. For det-rand, we currently only have trivial bounds: the lower bound from rand-rand and the upper bound from det-det.

---

### Official Review · Reviewer_YmDb · 2025-06-27

**Clarity:** 3
**Significance:** 2
**Originality:** 3
**Rating:** 4
**Confidence:** 4

**Summary:**

The paper studies distributed voting systems, where n voters are partitioned into k groups. Each group selects a local representative, and a final winner is chosen from the set of representatives (or all candidates). The key measure of interest is "distortion," which quantifies how much worse the outcome chosen by a voting rule (using only ordinal preferences) can be compared to the optimal outcome (knowing all voters' actual numerical costs or utilities) in a metric space. Distributed voting was first considered by Anshelevich et al. '22, and since then, only deterministic voting rules have been explored. This paper is the first to explore randomized voting rules in the distributed setup, though it is worth noting that randomized voting mechanisms in the distortion framework (standard non-distributed setting) have already been explored and are known to provide a better distortion bound compared to deterministic ones.

The current paper mostly focuses on the randomized-deterministic and randomized-randomized mechanisms. For rand-det, the authors provide matching (upper and lower) bounds for all four (avg-avg, avg-max, max-avg, and max-max) cases, while on rand-rand, they provide matching 3-distortion bounds for max-avg and max-max, and near-matching bounds for the other two cases. Further, the authors also manage to strengthen the lower bound of det-det mechanisms in the max-avg case, provide a tight 5-distortion lower bound, and improve the upper bounds of avg-max (10 to 7) and max-max case (5 to 3). In the end, the authors focus on the Euclidean metric and provide a lower bound on distortion for both rand-rand and rand-det mechanisms in the avg-avg case.

**Questions:**

Nothing specific

**Ethical Concerns:**

["NO or VERY MINOR ethics concerns only"]

**Final Justification:**

I did not have any major concern; on the other hand, I do not find the results or techniques super exciting (rebuttal does not give any additional information), and thus I am keeping the score as it is.

**Limitations:**

yes

**Quality:**

3

**Strengths And Weaknesses:**

This paper shows a host of new results on distributed voting mechanisms and shows many interesting tight bounds. One can say that a small weakness is that this paper does not introduce any new interesting techniques. Upper bound results, though interesting (some improve the known bounds, such as how to merge distortion bounds of outer and inner mechanisms to get a better final bound), are pretty simple. However, the lower bound constructions are clever, albeit with their simplicity.

---

> ### Author Rebuttal · Authors · 2025-07-31
>
> Thank you for taking the time to review our paper and provide sincere comments. Below, we address the technical comments.
>
> >**This paper shows a host of new results on distributed voting mechanisms and shows many interesting tight bounds. One can say that a small weakness is that this paper does not introduce any new interesting techniques. Upper bound results, though interesting (some improve the known bounds, such as how to merge distortion bounds of outer and inner mechanisms to get a better final bound), are pretty simple. However, the lower bound constructions are clever, albeit with their simplicity.**
>
> We agree that most the mechanisms we propose are simple mechanisms; however, analyzing certain settings, such as the average-average objective are very challenging. Additionally, the fact that such simple rules achieve tight distortion bounds is both intriguing and noteworthy. We also believe that the concepts of bias tournament and shortest-path metrics introduce valuable new tools for establishing lower bounds in various scenarios, which could be explored further in future work.

---

> > ### Comment · Reviewer_YmDb · 2025-08-05
> >
> > Thanks for your comment. I do not have any major concerns. It would really be good to know whether tools/techniques, like the bias tournaments or the shortest-path metrics, could be used for other scenarios. Hope they will find other applications in future works.

---

### Official Review · Reviewer_6jcC · 2025-07-03

**Clarity:** 3
**Significance:** 3
**Originality:** 3
**Rating:** 5
**Confidence:** 3

**Summary:**

This paper studies metric distortion in distributed voting systems, where voters are split into groups, each choosing a representative. The final winner is determined from these representatives. It focuses on four common cost objectives: avg-avg, avg-max, max-avg, and max-max, and provides tight upper and lower bounds for distortion with respect to deterministic and randomized mechanisms in general and line metric spaces. Some applications for this problem include certain elections, such as the US presidential election.

The authors use methods such as bias tournament constructions and shortest path distances in order to derive these bounds. Some of the existing bounds are improved, and most of the developed bounds are tight.

**Questions:**

1. This is mostly a well-written paper; however, it would be better to explain how distortion can be used in the real world.
2. Distortion seems to be a worst-case measure of sorts. It might help to include experiments to show how much distortion occurs in practice.

**Ethical Concerns:**

["NO or VERY MINOR ethics concerns only"]

**Final Justification:**

I am satisfied with the response and have no further questions.

**Limitations:**

Since this is a theoretical paper, it is not expected to consider the societal impact. However, is there a potential use case where the knowledge of distortion can be used to manipulate elections?

**Quality:**

3

**Strengths And Weaknesses:**

Strengths:
1. The paper is well written and includes the necessary details, while leaving the details to the appendix.
2. The problem in question regarding studying distortion bounds is an important one, and the paper provides tight upper and lower bounds for an extensive number of deterministic and randomized mechanisms.
3. The methods used, such as bias tournament and shortest path distances, in the proofs are interesting.

Weakness:
1. There is a lack of empirical results, and real-world applications could be better explained.

---

> ### Author Rebuttal · Authors · 2025-07-31
>
> Thank you for taking the time to review our paper and provide helpful suggestions. Below, we address the technical comments.
>
>
> >**Weakness: There is a lack of empirical results, and real-world applications could be better explained.**
>
> We acknowledge the importance of bridging theory and practice. The current version of the paper includes a wide range of results, and the conference page limit prevented us from including additional material. In the final version, we will clarify connections to real-world applications, such as federated elections (e.g., U.S. Electoral College, Indian Presidential Elections, and European Union Council Voting).
>
> >**Question1: It would be better to explain how distortion can be used in the real world.**
>
> Thanks for the suggestion; we will place more emphasis on real-world applications in the final version. Distortion is a well-established efficiency measure, meaningful in any decision-making context with limited information—such as voting and other real-world applications including matching, participatory budgeting, facility location, and so on.
>
> >**Question2: Distortion seems to be a worst-case measure of sorts. It might help to include experiments to show how much distortion occurs in practice.**
>
> Yes, distortion is commonly used as a worst-case benchmark, which can be overly pessimistic, as such extremes may not occur in real-world scenarios. It is therefore reasonable to consider less pessimistic settings, such as average-case analyses, which have been explored in prior work [1,2]. Adding an experimental component is a valuable suggestion. While it may be difficult to include in the current version due to space constraints, we will consider it for a more complete version, such as a future journal article. Thank you.
>
> >**Limitation: Since this is a theoretical paper, it is not expected to consider the societal impact. However, is there a potential use case where the knowledge of distortion can be used to manipulate elections?**
>
> Indeed, distortion depends on the voting rule itself, so it is not possible to determine the distortion before the rule is specified. Furthermore, since distortion is defined based on worst-case scenarios, it does not necessarily provide voters with useful information to strategically manipulate the election.
>
>
> References:
>
> [1]   Caragiannis, Fehrs. Beyond the worst case: Distortion in impartial culture electorates, 2023.
>
> [2]   Boutilier, Caragiannis, Haber, Lu, Procaccia, and Sheffet. Optimal social choice functions: A utilitarian view, 2012.

---

> > ### Comment · Reviewer_6jcC · 2025-08-07
> >
> > I am satisfied with the response and have no further questions.

---

### Official Review · Reviewer_LVHh · 2025-07-16

**Clarity:** 3
**Significance:** 3
**Originality:** 3
**Rating:** 5
**Confidence:** 4

**Summary:**

The paper studies metric distortion in distributed voting, where (1) voters are divided into groups, (2) each group selects a local winner, and (3) the group winners then select a final winner. The authors consider four social cost objectives and present improved upper and lower bounds on the distortion of both deterministic and randomized voting rules.

**Questions:**

The authors are encouraged but not required to answer the questions mentioned in “Strengths & Weaknesses” above.

**Ethical Concerns:**

["NO or VERY MINOR ethics concerns only"]

**Final Justification:**

Most (if not all) of my comments are constructive suggestions. The authors responded that they would revise the paper accordingly. A few short and clear points that justify my rating were provided under "Strengths and Weaknesses."

**Limitations:**

yes

**Quality:**

4

**Strengths And Weaknesses:**

Strengths
- The paper studies voting rules quantitatively through the lens of metric distortion, which complements the traditional axiomatic approach. There have been many theoretical advances in this area recently.
- It is natural to study the distortion of voting rules in the distributed setting, which could be of practical relevance in the long run. The presentation of the paper is clear.
- The paper presents solid technical results. The authors close the gap in worst-case distortion for the max-avg and max-max social objective for deterministic rules (and reduce the gap for avg-max), establish tight bounds in 6 out of 8 randomized settings they consider (and give nearly tight bounds in the 2 remaining cases).

Weaknesses
- The paper has a wide range of results but can read like a Cartesian product of different settings: 4 objectives, 3 randomness models (where is det-rand? I am not advocating for adding it), and various metric spaces. This can seem tedious to readers unfamiliar with the area. The paper could be strengthened by emphasizing which results are most novel or challenging to prove, and what ideas the authors propose (e.g., Bias Tournament) that are missing from prior work. How much are the authors' proofs inspired by that in Kizilkaya-Kempe?
- The authors could consider splitting Table 1 into two tables (one for deterministic and one for randomized) and/or adding the avg-avg row in det-det to provide a complete overview. (Are the upper and lower bounds of the avg-avg deterministic case still $7$ and $11$?). The paper should define $n^*$ in Table 1 (in addition to Theorem 4.4). The phrase "tight bounds" in the title may be a bit of an overstatement as some bounds are not tight, perhaps "tighter bounds"?

Typos
- Line 78: $\alpha$ is not defined yet, and should it be $\alpha+2$ instead of $5$?
- Line 330 and 955: "an unanimous"
- Line 347: "can not"
- Line 951: "these information"
- Line 960: "Observation A.11" missing brackets in the long equation

---

> ### Author Rebuttal · Authors · 2025-07-31
>
> Thank you for taking the time to review our paper and provide thoughtful comments. Below, we address the technical comments.
> ***
>  > **The paper has a wide range of results but can read like a Cartesian product of different settings: 4 objectives, 3 randomness models (where is det-rand? I am not advocating for adding it), and various metric spaces. This can seem tedious to readers unfamiliar with the area.**
>
> We acknowledge that some parts of the paper may feel dense to readers less familiar with the field. Our goal was to provide a comprehensive view of the problem, which, due to space constraints, may have led to a quicker transition into technical material than ideal. In the final version, we will ensure smoother transitions, clearer emphasis on the main ideas, and an overall more accessible presentation.
>
> >**Where is det-rand? I am not advocating for adding it.**
>
>  The randomized nature of in-group selection limits the direct applicability of techniques such as the Bias Tournament. Due to time and space constraints, we focused on the det-det, rand-rand, and rand-det models in this work. For det-rand, we currently only have trivial bounds: the lower bound from rand-rand and the upper bound from det-det. Analyzing the det-rand model is indeed a promising direction for future work.
>
> >**The paper could be strengthened by emphasizing which results are most novel or challenging to prove, and what ideas the authors propose (e.g., Bias Tournament) that are missing from prior work.**
>
> Thank you for the helpful points. In the final version, we will place more emphasis on which results are the most novel and technically challenging, as well as the new ideas introduced in the paper.
> For the upper bounds, the mechanisms in the same class are structurally simple but require different and nontrivial analyses for different cost functions. Among these, the upper bounds for the avg-avg objective in the rand-det and rand-rand settings are the most challenging  to obtain.
> For the lower bounds, we introduce several new ideas that we believe will be useful for future work. One is the Bias Tournament technique, which offers a flexible tool for analyzing distortion. Another is modeling the metric space via shortest paths in a graph. Additionally, our Euclidean lower bounds rely on a new construction that places voters and candidates on a hyper-simplex, which may offer further insight into geometric aspects of distortion.
> We will make sure to highlight these aspects more clearly in the final version to better convey the novelty and significance of our contributions.
>
> >**How much are the authors' proofs inspired by that in Kizilkaya-Kempe?**
>
> In most of the deterministic parts of our mechanisms, we can employ either the plurality-veto or plurality-matching rule, which achieves the distortion of 3 in the metric setting. While there may arise similarities with prior work due to the problem's nature, our paper was developed independently of Kizilkaya–Kempe [2,3].
>
> >**The authors could consider splitting Table 1 into two tables (one for deterministic and one for randomized) and/or adding the avg-avg row in det-det to provide a complete overview. (Are the upper and lower bounds of the avg-avg deterministic case still 7 and 11?) The paper should be defined n\* in Table 1 (in addition to Theorem 4.4).**
>
> Due to page limitations, we consolidated all of our results into a single table. We will do our best to address this concern in the final version. For the avg-avg objective under the det-det model, to the best of our knowledge, the best-known bounds are 7 and 11, as established in [1], and we will include these in the final version of the table to provide a more complete overview. We also apologize for missing the definition of n* in the tables and will address this by clearly defining it.
>
> >**The phrase "tight bounds" in the title may be a bit of an overstatement as some bounds are not tight, perhaps "tighter bounds"?**
>
> Since the majority of our results—6 out of 8 in the randomized setting (rand-rand and rand-det) and 2 out of 3 in the deterministic case (det-det)—are indeed tight, we felt the current title was justified. However, we are open to revising it to something like “Bounds on the Distortion” to more accurately reflect the scope of the results.
>
> >**Line 78: ɑ is not defined yet, and should it be ɑ+2 instead of 5?**
>
> Here, we mean that “any deterministic in-group rule with distortion at most \alpha (for \alpha>= 3), followed by Random Dictatorship, achieves an overall distortion of \alpha+2. Since the best possible value for α is 3, this yields an upper bound of 5”. In the final version, we'll write this part more carefully.
>
> **Typos:**
> Thanks; in the final version, we’ll address the other typos you pointed out.
>
> **References:**
>
> [1]   Anshelevich, Filos-Ratsikas, and Voudouris, “The distortion of distributed metric social choice,” Artificial intelligence, 2022.
>
> [2]   Kizilkaya and Kempe, “Plurality veto: A simple voting rule achieving optimal metric distortion,” IJCAI 2022.
>
> [3]   Kizilkaya and Kempe, “Generalized veto core and a practical voting rule with optimal metric distortion,” EC 2023.

---

> > ### Comment · Reviewer_LVHh · 2025-08-05
> >
> > I thank the authors for their response. I have no further comments or questions at this point.

---

### Decision · Program_Chairs · 2025-09-17

**Decision:**

Accept (poster)

**Comment:**

The paper studies distributed voting systems, where voters are partitioned into groups. A representative is chosen by each group, and the final winner is selected from these representatives (or all candidates). The key measure of interest is "distortion."

The reviewers were unanimous in praise of the strength of the results and of the writing quality. The proofs of the lower bounds introduce new techniques which may be useful in the future.

There were a number of minor issues which were pointed out by the reviewers, and we would request the authors to address them in the final draft.